# Mild replication stress causes chromosome mis-segregation via premature centriole disengagement

Therese Wilhelm[1,4,6], Anna-Maria Olziersky[1,6], Daniela Harry[1], Filipe De Sousa[1], Helène Vassal[1,2], Anja Eskat[1,5] & Patrick Meraldi[1,3]

Replication stress, a hallmark of cancerous and pre-cancerous lesions, is linked to structural chromosomal aberrations. Recent studies demonstrated that it could also lead to numerical chromosomal instability (CIN). The mechanism, however, remains elusive. Here, we show that inducing replication stress in non-cancerous cells stabilizes spindle microtubules and favours premature centriole disengagement, causing transient multipolar spindles that lead to lagging chromosomes and micronuclei. Premature centriole disengagement depends on the G2 activity of the Cdk, Plk1 and ATR kinases, implying a DNA-damage induced deregulation of the centrosome cycle. Premature centriole disengagement also occurs spontaneously in some CIN+ cancer cell lines and can be suppressed by attenuating replication stress. Finally, we show that replication stress potentiates the effect of the chemotherapeutic agent taxol, by increasing the incidence of multipolar cell divisions. We postulate that replication stress in cancer cells induces numerical CIN via transient multipolar spindles caused by premature centriole disengagement.

[1] Department of Cell Physiology and Metabolism, University of Geneva, 1211 Geneva 4, Switzerland. [2] National Institute of Applied Sciences, Villeurbanne 69621, France. [3] Translational Research Centre in Onco-hematology, University of Geneva, 1211 Geneva 4, Switzerland. [4] Present address: Department of Genetic Stability and Oncogenesis, Institut Gustave Roussy, CNRS UMR8200, 94805 Villejuif, France. [5] Present address: Clinical Trials Center, University Hospital Zurich, 8091 Zurich, Switzerland. [6] These authors contributed equally: Therese Wilhelm, Anna-Maria Olziersky. Correspondence and requests for materials should be addressed to T.W. (email: Therese.Wilhelm@gustaveroussy.fr) or to P.M. (email: Patrick.Meraldi@unige.ch)

Chromosomal instability (CIN) is a hallmark of cancer that correlates with poor prognosis[1]. It refers to the increased rate through which chromosomes undergo numerical or structural changes in a cell. Studies in colorectal cancer cells have shown that CIN is already present in early adenoma[2], suggesting that it is an early event during tumorigenesis. CIN can be structural or numerical. Structural CIN, in which a chromosome part is lost or attached to another chromosome, can be due to chromosome breakages or chromosome rearrangements. Unreplicated chromosome parts or dicentric chromosomes can lead to DNA bridges between the two DNA masses that are prone to chromosome breakages[3]. Numerical CIN is thought to arise as a consequence of mitotic dysfunctions. The most frequent cause is lagging chromosomes in anaphase that are due to merotelic kinetochore-microtubule attachments[4,5], in which the kinetochore on one sister chromatid binds to microtubules emanating from both spindle poles[6]. If merotely is not corrected, this sister chromatid is subjected to tug-of-war in anaphase, creating the potential for chromosome mis-segregation. Even when such chromosomes are segregated to the correct daughter cell, as is often the case late in anaphase[7], frequently it will become a micronucleus prone to chromothrypsis[8]. Importantly, micronuclei disintegration and release of DNA into the cytoplasm can activate inflammatory signalling and drive metastasis[9].

The cellular causes for lagging chromosomes are thought to be mitotic defects[4]: non-cohesed or unreplicated single chromatids will easily form merotelic attachments, as their single kinetochore can be reached by microtubules from both spindle poles[10,11]; transient monopolar spindles or bipolar spindles in which the two poles are not perfectly juxtaposed will promote merotely due to geometrical reasons; similarly, multipolar spindles, which can arise through defects in centrosome number or structure[12] form numerous merotelic attachments before clustering back into a bipolar configuration prior anaphase[13,14]. The correction of merotely depends on the dynamic instability of microtubules, as it allows the stochastic release of incorrect attachments[15]. Consequently, increasing microtubule stability also leads to more lagging chromosomes as it prevents error correction[16]. Interestingly, cancerous CIN+ cells have an intrinsically higher microtubule stability than non-cancerous retinal pigment epithelial cells, resulting in more persistent merotelic kinetochore-microtubule attachments[17]. Nevertheless, the exact molecular cause for the increased microtubule stability and numerical CIN remains unclear, as mutations in mitotic genes are rare in cancer[18].

One surprising potential cause for numerical aneuploidy is DNA replication stress[19]. Any obstacle that perturbs DNA replication and prevents cells from completing genome duplication before mitosis is considered replication stress[20]. It is a frequent characteristic of cancers and pre-cancerous lesions that has been associated to structural CIN. Many cancer cells intrinsically harbour replication stress due to oncogene activation (e.g. Myc, Cyclin E amplification), compromised DNA repair or chromosomal imbalances and the ensuing proteotoxic stress[21–23]. Recent studies demonstrated that colorectal cancer cells or artificially generated polyploid cells were also prone to numerical CIN as induction of replication stress with Aphidicolin increased changes in chromosome numbers[19,24]. The mechanisms by which replication stress affects chromosome segregation remain, however, unclear. Indeed, different types of replication stress might have a broad spectrum of repercussions in mitosis. Replication stress induced by high doses of hydroxyurea (HU) or cyclin E over-expression provokes chromosome breaks due to transcription/replication conflicts[25,26]. Cancerous cells counterbalance such strong stress by inter-genic origin firing[26] or up-regulation of interphase or mitotic DNA repair[27,28]. Strong DNA replication stress will also lead to an extended S-phase delay that favours centrosome overduplication, leading to supernumerary centrosomes[29]. It will, however, strongly reduce mitotic entry in checkpoint proficient non-cancerous cells, limiting its impact[30]. In contrast, mild replication stress constitute a more severe threat, as interphase checkpoints may fail to detect it, allowing its propagation in non-cancerous cells[31]. Mild stress induced by low doses of the DNA polymerase inhibitor Aphidicolin, is associated with mild interphase delay and mitotic chromosome breaks[32], revealed as lagging acentric chromatin fragments in anaphase and ultrafine bridges[33]. Mild replication stress has also been reported to lead to a prolonged mitosis in p53 mutated Chinese hamster ovary cells concomitant with the appearance of extra mitotic spindle poles and DNA bridges[34].

Here we identify premature centriole disengagement as a key mechanism by which replication stress can induce numerical CIN in non-cancerous, checkpoint proficient human cell lines. This causes transient multipolar spindles, leading to lagging chromosomes in anaphase and micronuclei formation. Moreover, we show that mild replication stress in un-transformed cells is associated to microtubule stabilization, which further favours premature centriole disengagement. We postulate that premature centriole disengagement is at the origin of numerical CIN in cells with replication stress. Consistent with this hypothesis, we identify several colorectal and breast cancer cell lines that display replication stress-dependent premature centriole disengagements. Finally, we demonstrate that replication stress potentiates the effect of taxol, a chemotherapeutic agent that kills cancer cells via multipolar cell divisions[35].

## Results

**Replication stress induces whole chromosome mis-segregation.** To investigate the potential causes for chromosome segregation defects and numerical aneuploidy after mild replication stress, we worked with non-cancerous, chromosomally stable RPE1 cells immortalized with human telomerase. These cells have functional cell-cycle checkpoints and a low basic incidence of chromosome segregation errors. To induce replication stress, we applied for 16 h low doses (200 or 400 nM) of Aphidicolin, a DNA polymerase inhibitor that mimics replication stress. 200 or 400 nM Aphidicolin induced mild replication stress in RPE1 cells, as immunofluorescence revealed a dose-dependent, slight increase in the percentage of cells that stained positive for phospho-γH2AX (Fig. 1a, b), and an increase in mitotic chromosome breaks, as visualized by metaphase chromosome spreads (Fig. 1c, d). Low doses of Aphidicolin did, however, not induce extensive interphase DNA damage, since the percentage of cells harbouring foci of the DNA damage marker 53BP1 was unchanged (Fig. 1a, e). In contrast, 1 mM HU, a condition that induces high levels of replication stress and DNA damage, strongly increased both the percentage of 53BP1- and phospho-γH2AX-positive cells (Fig. 1a, e).

To evaluate the effect of replication stress on chromosome segregation we monitored RPE1 cells expressing EB3-GFP (plus end microtubule marker) and H2B-RFP (chromatin marker) for 12 h by time-lapse imaging. In particular, we quantified whether cells segregated their chromosomes without visible defects, with lagging DNA (H2B positive DNA left behind the two DNA masses in anaphase) or DNA bridges (thin DNA thread connecting the two DNA masses). Untreated cells displayed few DNA bridges (2.9 %) and even fewer lagging DNA (<1%; Fig. 1f). Applying 200 or 400 nM Aphidicolin did not increase the incidence of DNA bridges; rather it led to a dose-dependent increase in lagging DNA (4.5% for 200 and 10.7% for 400 nM Aphidicolin), suggesting that mild replication stress may lead to erroneous chromosome segregation in RPE1 cells (Fig. 1f, g).

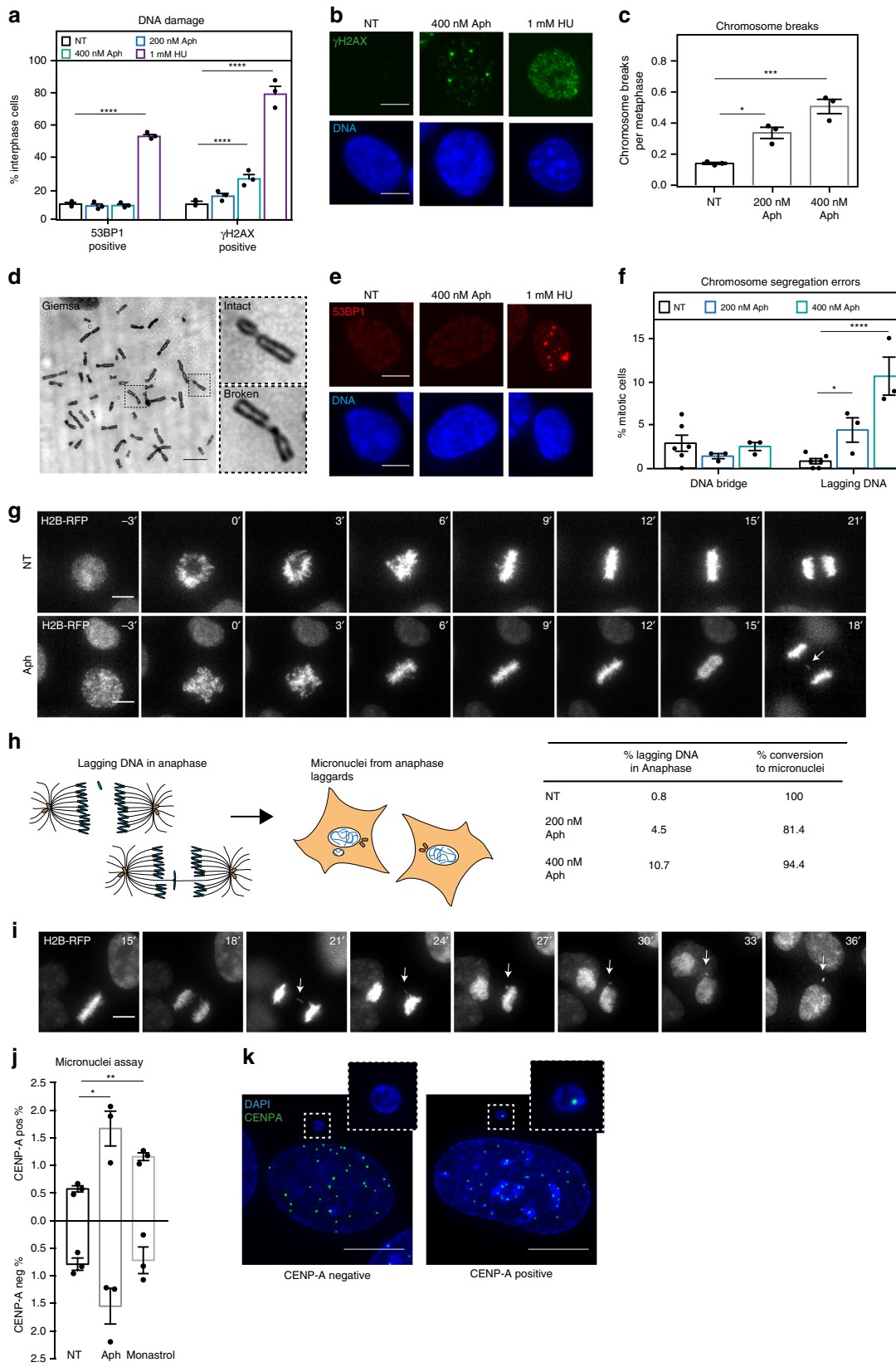

Consistently, most (80–90%) lagging DNA ended up in micronuclei (Fig. 1h, i).

The presence of lagging DNA could be due to mis-segregation of whole chromosomes or the presence of acentric chromosome fragments. We analysed micronuclei for the presence of the centromere marker CENP-A, since mis-segregation of entire or near-entire chromosomes requires centromeres. In non-treated RPE1 we found that both CENP-A positive micronuclei (0.6%) and CENP-A negative micronuclei (0.8%) were rare (Fig. 1j). A transient treatment with monastrol, an inhibitor of the spindle

**Fig. 1** Mild replication stress induces whole chromosome mis-segregation. **a** Mean percentage of γH2AX and 53BP1 positive RPE1 cells. N = 3 independent experiments examining NT = 909, 200 nM Aph = 963, 400 nM Aph = 971 and 1 mM HU = 637 cells; p < 0.0001 in two-way Anova test (γH2AX: NT vs. 400 nM Aph p < 0.0001; NT vs. HU p < 0.0001; 53BP1: NT vs. HU p < 0.0001). **b** Representatives images of cells stained for γH2AX and DNA after indicated treatments. All scale bars in Figure = 10 μm. **c** Number of metaphase chromosome breaks. N = 3 examining NT = 222 and Aph = 212 metaphase spreads; p = 0.0008 in one-way Anova test (NT vs. 200 nM Aph p = 0.0114; NT vs. 400 nM Aph p = 0.0005). **d** Representative image of a Giemsa stained metaphase after a 400 nM Aph treatment; insets show an intact and broken chromosome. **e** Representative images of RPE1 cells stained for 53BP1 and DNA after indicated treatments. **f** Percentage of chromosome segregation errors as quantified from time-lapse imaging experiments of RPE1 H2B-mCherry/EB3-GFP cells. N = 3 (Aph) and 6 (NT) examining NT = 551, 200 nM Aph = 423 and 400 nM Aph = 195 anaphases; p = 0.0003 in two-way Anova test (lagging DNA: NT vs. 200 nM Aph p = 0.04; NT vs. 400 nM Aph p = 0.0001). **g** Representative time-lapse sequence after indicated treatments quantified in (**f**). Nuclear envelope breakdown (NEBD) is t = 0. White arrow indicates lagging DNA. **h** Conversion rate of lagging DNA to micronuclei according to the left schematic (created by TW). **i** Representative time-lapse sequence of cells forming a micronucleus from persistent lagging DNA (white arrow). NEBD is t = 0. **j** Percentage of interphase cells containing CENP-A-negative and CENP-A-positive micronuclei after indicated treatments. N = 3 examining NT = 2928, Aph = 2421 and Monastrol = 2653. CENP-A positive MN: NT vs. Aph p = 0.0269 in unpaired t-test; NT vs monastrol p = 0.0029. **k** Representatives image of cells with micronuclei containing (right) or not (left) CENP-A signal; error bars indicate sem. Source data are provided as a Source Data file

motor Eg5, led to a specific two-fold increase in CENP-A positive micronuclei, consistent with previous studies showing that the transient formation of a monopolar spindle favours mis-segregation of whole chromosomes[36]. In contrast, treating RPE1 cells with Aphidicolin resulted in an increase of both CENP-A positive (3-fold; 1.7%) and CENP-A negative micronuclei (2-fold, 1.6%; Fig. 1j). This implied, that replication stress not only led to acentric chromosome fragments, as seen in the metaphase spread, but also to mis-segregation of whole chromosomes. Our quantification specifically suggested that roughly half the lagging DNA correspond to mis-segregating chromosomes.

**Mild replication stress stabilizes microtubules**. Most lagging whole chromosomes in anaphase results from merotelic-kinetochore microtubule attachments[5]. Merotely can arise as a consequence of centromere cohesion defects[37], increased microtubule stability[15], defects in spindle assembly[38–40], or multipolar spindles[13,14]. We first tested whether low doses of Aphidicolin might lead to chromosome cohesion defects. Consistent with a previous study[19], no increase in cohesion defects could be observed in metaphase spreads (Supplementary Fig. 1a).

To investigate whether mild replication stress stabilizes microtubules, we treated cells with cold, since cold-resistance reflects the stability of spindle microtubules. Cells were put on ice for 12 min, stained for α-tubulin, and assigned to three classes based on fluorescence microscopy: metaphase cells with intact spindles (Class 1), metaphase cells with mitotic spindles having partially lost their integrity (Class 2) and metaphase cells in which the mitotic spindle was no longer recognizable (Class 3; Fig. 2a). Treating RPE1 cells with 400 nM Aphidicolin shifted the distribution towards Class 1 (Fig. 2a, b), suggesting that mild replication stress stabilizes spindle microtubules.

To corroborate these results we established an assay that directly monitors microtubule spindle stability. We labelled the mitotic spindles of cells arrested in metaphase with the proteasome inhibitor MG132 with the live cell imaging dye SIR-tubulin and applied a short pulse of 200 nM nocodazole to induce microtubule depolymerization. Immediately after, we recorded the decay of the spindle signal by live cell imaging (Fig. 2c). This assay provides an excellent signal-to-noise ratio, since SIR-tubulin is a docetaxel-based fluorescent dye that labels only polymerized microtubules and not soluble tubulin dimers. For validation purposes, we compared the spindle behaviour in cells treated with a control siRNA or cells depleted of the microtubule depolymerases MCAK and Kif2a. Consistent with previous studies[41], MCAK/Kif2a depletion significantly stabilized spindle microtubules (Fig. 2c, d). We next compared the behaviour of mitotic spindles with or without replication stress

and found significant microtubule stabilization in cells treated with 400 nM Aphidicolin, confirming our cold stable assay results (Fig. 2e, f). To verify that the effects seen after Aphidicolin treatment were due to a perturbed DNA replication and not an off-target effect, we also treated RPE1 cells with Aphidicolin for only 1.5 h, and analysed microtubule stability in metaphase. Indeed, we reasoned that any cell in metaphase would have been already in G2 when we applied Aphidicolin, and thus should not suffer from replication stress. Consistent with our hypothesis, a short term Aphidicolin treatment did not affect microtubule stability (Supplementary Fig. 2a). Next we tested whether this phenotype could be also seen in other non-cancerous epithelial cells. When we treated MCF10A, a non-cancerous mammary epithelial cell line, with low doses of Aphidicolin, we found indeed a similar stabilization of microtubules (Fig. 2g, h). This suggested that that microtubule stabilization is a common response to Aphidicolin-induced replication stress in non-cancerous cells.

Finally, to test whether the elevated microtubule stability seen after mild replication stress contributes to chromosome mis-segregation, we combined a 16-hour treatment of 400 nM Aphidicolin with low doses of the microtubule-destabilizing drug nocodazole (10 ng/ml), and quantified the percentage of CENP-A-positive and negative micronuclei. We found that addition of nocodazole specifically reduced the percentage of CENP-A positive micronuclei, suggesting that replication stress-induced microtubule stabilization contributes to chromosome mis-segregation (Fig. 2i).

**Replication stress causes premature centriole disengagement.** Another potent driver of merotelic attachments are defects in spindle assembly and spindle architecture. We, therefore, re-analysed our live cell imaging movies, focusing on the spindle marker EB3-GFP. It revealed that mild replication stress was associated with a dose-dependent increase in multipolar spindles (Fig. 3a, b). Consistent with previous studies, nearly all multipolar spindles were transient in nature, as spindle poles clustered back into a bipolar configuration before anaphase. Previous studies have shown that transient multipolarity favours the formation of merotelic attachments resulting more frequently in lagging chromosomes[13,14]. Consistently, Aphidicolin-treated cells that formed transient multipolar spindles had a higher probability to display lagging DNA than those with pure bipolar spindles (31% vs. 5%; Fig. 3c), implying that multipolar spindles are a potent driver for lagging chromosomes. Nevertheless, when we excluded cells with transient multipolar spindles, we found that Aphidicolin-treated cells still had an incidence of lagging DNA that was an order of magnitude higher than non-treated cells (5%

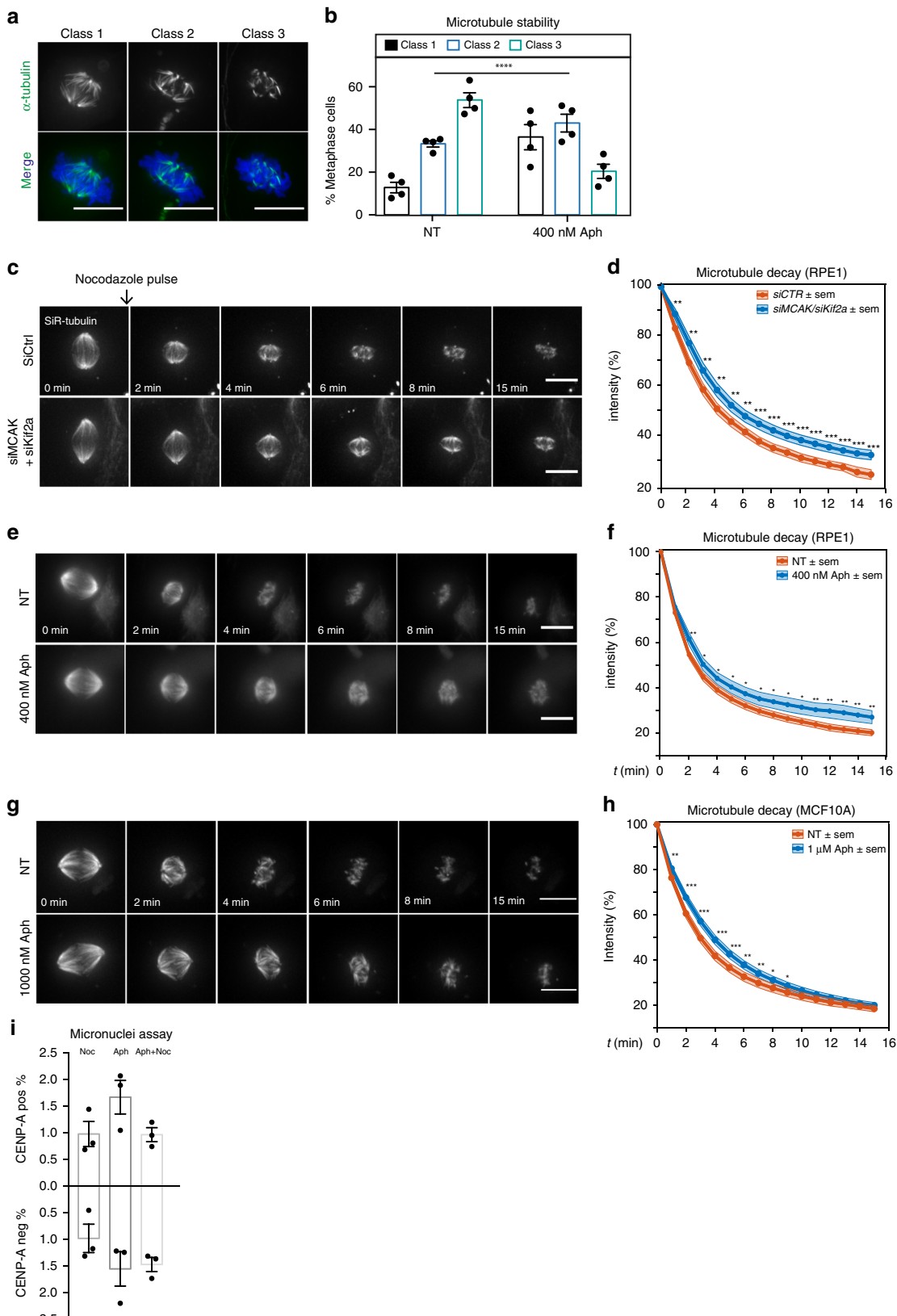

vs. 1%; Fig. 3c), suggesting that higher microtubule stability may also lead to lagging chromosomes on its own.

Next, we investigated what type of spindle multipolarity arises in Aphidicolin-treated cells. Known examples include multipolar spindles with extra centrosomes, e.g. due to centrosome over-

duplication or cytokinesis failure, multipolar spindles with acentriolar spindle poles, as observed after taxol treatment[42], or extra poles with single centrioles due to the splitting of the two centrioles within a centrosome (centriole disengagement)[12,43]. To differentiate between these possibilities, we quantified the

**Fig. 2** Mild replication stress stabilizes microtubules. **a** Representative images of cold- treated RPE1 centrin1-GFP cells stained with α-tubulin antibodies (green) and DAPI (blue). Cells were categorized into Class 1, 2 or 3 depending on the integrity of the mitotic spindle. All scale bars in Figure = 10 μm. **b** Mean percentages of class 1, class 2 and class 3 RPE1 centrin1-GFP cells; $N = 4$ examining in NT = 159 and Aph = 170 metaphases; $p < 0.0001$ in chi-square test. **c** Representative time-lapse images of metaphasic RPE1 centrin1-GFP cells stained with SiR-tubulin and treated with a nocodazole pulse at $t = 0$ after indicated treatments. **d** Quantification of the spindle intensity over time after a nocodazole pulse, as shown in (**c**), in metaphase RPE1 centrin1-GFP cells; $N = 33$ metaphases for siCTR and 31 for siMCAK/Kif2a. $*p < 0.05$; $**p < 0.01$; $***p < 0.001$ in a repeated measurement Anova test. **e** Representative time-lapse images of metaphasic RPE1 centrin1-GFP cells stained with SiR-tubulin and treated with a nocodazole pulse at $t = 0$ after indicated treatments. **f** Quantification of the spindle signal as in (**d**); $N = 32$ for NT and 31 for Aph. $*p < 0.05$; $**p < 0.01$; $***p < 0.001$ in a repeated measurement Anova test. **g** Representative time-lapse images of metaphasic MCF10A cells, stained with SiR-tubulin and treated with a nocodazole pulse at $t = 0$ after indicated treatments. **h** Quantification of the spindle signal as in (**d**) and (**f**); $N = 49$ metaphases/condition. $*p < 0.05$; $**p < 0.01$; $***p < 0.001$ in a repeated measurement Anova test. **i** Mean percentage of RPE1 cells containing CENP-A-negative and CENP-A-positive micronuclei, treated with 10 ng/μl Nocodazole, 400 nM Aphidicolin or both combined. $N = 3$ examining Noc = 3125, Aph = 2421 and Aph/Noc = 2397 interphase cells, error bars indicate sem. CENP-A positive MN: Noc vs. Aph $p = 0.1539$; Aph vs Aph/Noc $p = 0.0029$ in unpaired t-test. Source data are provided as a Source Data file

centriole numbers in extra spindle poles using RPE1 Centrin1-GFP (centriole marker) cells stained against the spindle pole marker γ-tubulin. In >90% of the Aphidicolin-treated RPE1-cells with multipolar spindles, we found single centrioles surrounded by γ-tubulin in the extra spindle poles, indicating that mild replication stress disengages centrioles (Fig. 3d, e). In contrast, short term Aphidicolin treatment did not lead to centriole disengagement (Supplementary Fig. 2b), indicating that Aphidicolin only affects mitotic centrosomes when it could impede DNA replication in the preceding interphase. Live cell imaging showed that mild replication stress did not change median mitotic timing by more than 3 min (Fig. 3f), which indicated that centriole disengagement was not caused by a prolonged mitotic duration, unlike what has been seen in other conditions[44]. Rather, centriole disengagement and transient multipolar spindles were already visible shortly after nuclear envelope breakdown in both fixed and live cells (Fig. 3b, e). This suggested that replication stress induces untimely disengagement of centrioles in early mitosis and not like in a normal cell cycle in late telophase. A similar multipolarity due to premature centriole disengagement was also found in MCF10A cells treated with Aphidicolin (Fig. 3g, h; note that a small subset of the multipolar spindles were also due to extra centrosomes in this case), indicating that this phenotype was not cell line specific, but more general to non-cancerous cells.

In a last step, we tested whether the elevated microtubule stability is linked to premature centriole disengagement, by combining the Aphidicolin-treatment with low doses of nocodazole (10 ng/ml). Centriole disengagement induced after 16 h Aphidicolin treatment was partially suppressed with a 16 h co-treatment with nocodazole, suggesting that microtubule stabilization contributes to the disengagement (Fig. 3h). It does, however, not suffice, since microtubule stabilization obtained after MCAK/Kif2a depletion did not induce premature centriole disengagement[41]. We conclude that replication stress leads to lagging chromosomes due to premature centriole disengagement and transient multipolar spindles that partially depend on microtubule stabilization.

**Mild replication stress acts via Cdk1, Plk1 and ATR in G2.** The fact that centrioles separated immediately at mitotic entry raised the possibility that centrosomes were too advanced in their cycle. Based on FACS cell cycle analysis and immunofluorescence staining with the G2 marker CENP-F, we found that 200 or 400 nM Aphidicolin led to fewer mitotic cells and a higher proportion of G2 cells (Fig. 4a; Supplementary Fig 1b, c and; Supplementary Fig. 3). This implied a prolonged G2 phase. Since replication stress induces a cell cycle delay via the ATR-Chk1 signalling axis, we next tested whether premature centriole disengagement also depended on ATR using 800 nM ETP-47474, an ATR inhibitor.

FACS analysis indicated that such a low dose of ETP-47474, only moderately changed cell cycle dynamics, without inducing a accelerated mitotic entry (Supplementary Fig. 4). At these doses it fully suppressed premature centriole disengagement, confirming that this phenotype depends on the ATR-Chk1 signalling axis (Fig. 4b). A further analysis of the mitotic phenotype was however not possible, since ATR inhibition also leads to severe chromosome segregation defects on its own[45].

To test whether a prolonged G2 alone induces premature centriole disengagement, RPE1 cells were treated for 16 h with a Cdk1 inhibitor (RO-3306), before release into mitosis. As previously shown[46], this treatment led to a strong enrichment of CENP-F positive G2 cells (Fig. 4a and Supplementary Fig. 3). After release it did, however, not induce premature centriole disengagement (Fig. 4c). As Cdk1 itself is implicated in centriole biology[47] and microtubule stability[48], we also combined Cdk1 inhibition with Aphidicolin treatment and found that a 16 h Cdk1 inhibition suppressed premature mitotic centriole disengagement in Aphidicolin-treated cells (Fig. 4c). This indicated that premature centriole disengagement depends on Cdk1 activity in G2. Strikingly, premature centriole disengagement was also suppressed with RO-3306 treatment in the last 90 min of the 16 h Aphidicolin treatment, indicating that Cdk1 activity is essential for centriole disengagement only briefly before mitosis (Fig. 4d). Given that mild replication stress also increased microtubule stability we also tested whether this phenotype depends on Cdk1 activity in G2. This was indeed the case, as a Cdk1 inhibition pulse suppressed the increase in spindle microtubule stability of Aphidicolin-treated cells (Fig. 4e, f). Importantly, in the absence of Aphidicolin treatment, 16 h Cdk1 inhibition and release did not change the stability of spindle microtubules (Fig. 4g). Finally, we also tested whether Cdk1 inhibition could also suppress the appearance of CENP-A positive micronuclei, and found that this was indeed the case (Fig. 4h). In contrast, Cdk1 inhibition did not affect the CENP-A negative acentric chromosomes, indicating that Cdk1 activity is specifically required for the mis-segregation of whole chromosomes.

One of the key downstream targets of Cdk1 in G2 is Bora, an activator of the Polo-like kinase Plk1[49]. Since Plk1 activity is a driver of centriole disengagement in telophase[50], we further tested whether premature centriole disengagement also depended on Plk1 activity. As full inhibition of Plk1 prevents mitotic entry[51], we used low doses of the Plk1 inhibitor BI2536, to partially inhibit Plk1 and allow mitotic entry (Supplementary Fig. 4), and found a partial, dose-dependent suppression of premature centriole disengagement (Fig. 4i). We conclude that premature centriole disengagement depends on both Cdk1 and Plk1 activity.

**Centriole disengagement is present in CIN+ cancer cells.** To test whether premature centriole disengagement in prometaphase

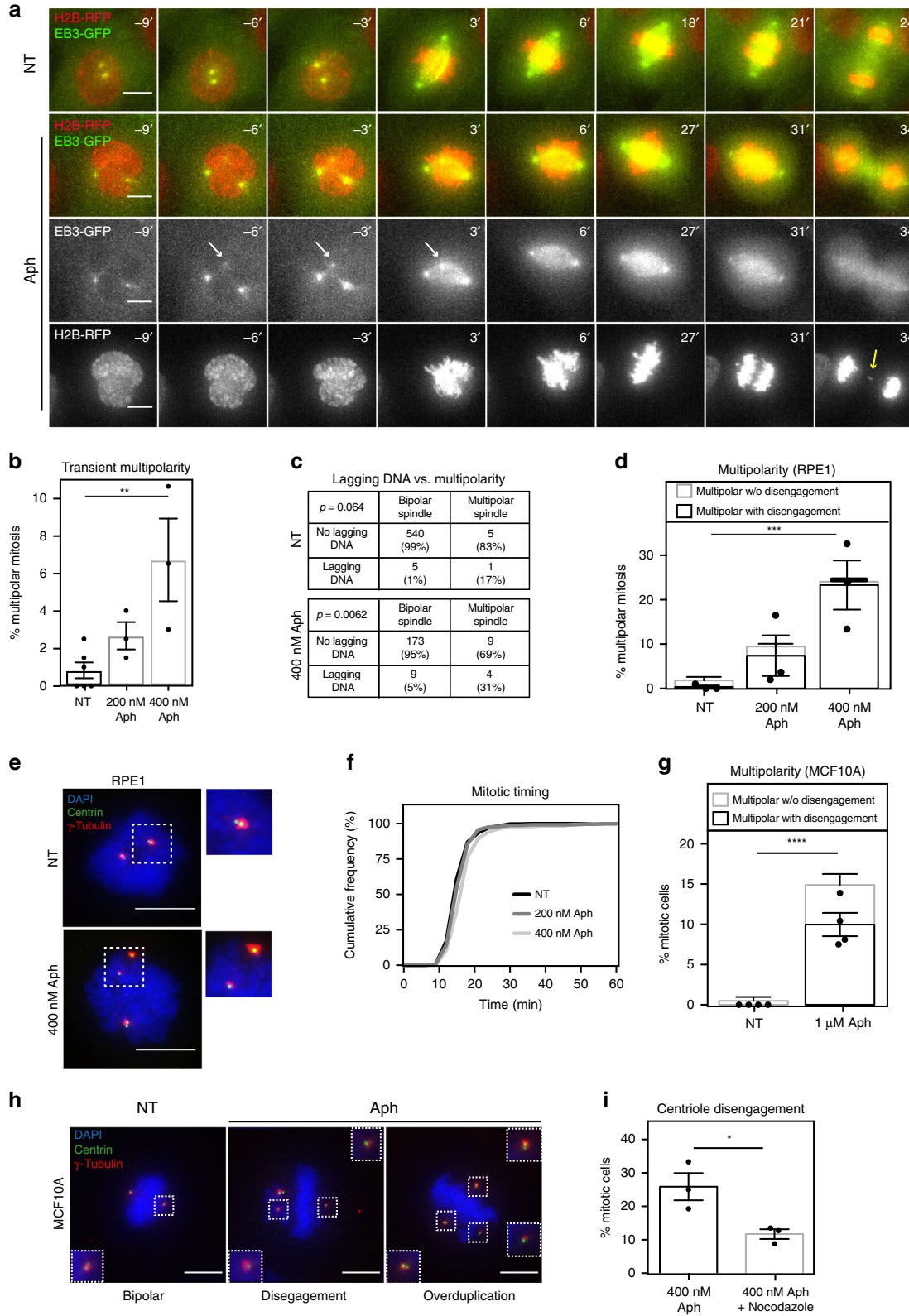

also arises in the context of endogenous replication stress in cancer cells, we analysed a panel of 7 colorectal and breast cancer cell lines that were either (3) CIN− or (4) CIN+. This panel included in particular colorectal CIN+ cancer cell lines HT29 and H747, in which numerical CIN has been linked to replication stress[19]. Immunofluorescence staining with antibodies against centrin and γ-tubulin revealed in the colorectal CIN+ cancer cell line HT29 and two CIN+ breast cancer cell lines HCC70 and HCC1187 > 5% of cells with multipolar spindles caused by premature centriole disengagement (Fig. 5a, b). On top, an equal number of multipolar spindles displayed a combination of disengaged centrioles and overduplicated centrosomes (Fig. 5b). In

**Fig. 3** Replication stress leads to transient spindle multipolarity. **a** Representatives time-lapse images of RPE1 H2B-mCherry/EB3-GFP cells after indicated treatments showing a normal (top) or transient multipolar spindle (white arrow) followed by a lagging chromosome in anaphase (yellow arrow); NEBD is set as $t = 0$; All scale bars in Figure $= 10$ μm. **b** Mean percentage of transient multipolar spindles quantified from time-lapse images as shown in (**a**); $N = 3$ for Aph and 6 for NT examining NT $= 551$, 200 nM Aph $= 423$ and 400 nM Aph $= 195$ anaphases; $p = 0.009$ in one-way Anova (NT vs. 400 nM Aph; $p = 0.005$). **c** Number of NT or Aphidicolin-treated RPE1 H2B-mCherry/EB3-GFP cells with or without lagging DNA depending on whether they have formed a multipolar spindle or not; same $N$ as in **b** (indicated $p$-values were calculated with Fischer's test). **d** Mean percentages of multipolar mitoses with or without centriole disengagement in RPE1 Centrin1-GFP cells; $N = 3$; $n = 46$–190; $p = 0.03$ in two-way Anova (Multipolar with disengagement: NT vs. 400 nM Aph $p = 0.0003$). **e** Representative images of prometaphase RPE1 centrin1-GFP (green) cells stained with γ-tubulin (pericentriolar material marker; red) and DAPI. Represented is a non-treated cell with engaged centrioles (top) and an Aphidicolin-treated cell with disengaged centrioles (bottom). **f** Cumulative frequency plot of the time between nuclear envelope breakdown ($t = 0$) and anaphase onset in RPE1 H2B-mCherry/EB3-GFP cells after indicated treatments; same $N$ and number of cells as in (**b**). **g** Mean percentages of multipolar mitoses with or without premature centriole disengagement in MCF10A cells; $N = 4$ examining NT $= 189$ and Aph $= 161$ mitoses; $p < 0.0001$ in two-way Anova (multipolarity with disengagement $p < 0.0001$, multipolarity without disengagement $p = 0.0201$). **h** Representative images of prometaphase MCF10A cells stained with antibodies against γ-tubulin (red), centrin (centriole marker; green) and DAPI. Shown is a non-treated cell with engaged centrioles (left), and Aphidicolin-treated cells with disengaged centrioles (middle) or centriole overduplication (right). **i** Mean percentage of RPE1 centrin1-GFP cells with dis-engaged centrioles; $N = 3$ examining Aph $= 168$ and Aph/Noc $= 118$ mitoses; $p = 0.03$ in unpaired $t$-test. Error bars indicate sem

contrast, all three CIN− cancer cell lines had either no or very low levels of premature centriole disengagement, while, the fourth CIN+ cell line, H747, displayed mostly multipolar spindles with overduplicated centrosomes.

To test whether the premature centriole disengagement resulted from the endogenous replication stress, we next supplemented HT29, HCC70 and HCC1187 with nucleosides, a condition that partially rescues replication stress[19]. This treatment reduced centriole disengagement in all three cell lines, indicating that endogenous replication stress is also at the basis of this phenotype in a cancer cell context (Fig. 5c). Finally, we tested whether endogenous replication stress was also associated to microtubule stabilization in CIN+ cancer cells. Nucleoside treatment indeed reduced microtubule stability in HT29 cells (Fig. 5d), indicating a link in this particular cell line. This was, however, not a general phenomenon, since a similar treatment did not affect microtubule stability in HCC70 or HCC1187 cells (Fig. 5e, f). We conclude that endogenous replication stress in cancer cells is associated to premature centriole disengagement, but that it is not necessarily associated to microtubule stabilization.

**Cells with replication stress are more sensitive to taxol.** Finally, we tested whether the increased microtubule stability and the propensity to form transient multipolar spindles in Aphidicolin-treated RPE1 sensitizes them for taxol. We specifically tested by live cell imaging whether mild replication stress exacerbates the ability of taxol to kill cancer cells via multipolar cell division[35]. We treated RPE1 H2B-mCherry/EB3-GFP cells either with 6 nM taxol alone or in combination with 400 nM Aphidicolin and measured the frequency of multipolar divisions (Fig. 6a). Even though taxol-treatment led to varying incidence of multipolar spindles (most likely reflecting its heterogeneous uptake[52]), we observed that mild replication stress increased the incidence of multipolar cell divisions in 7 out of 8 independent experiments (Fig. 6a, b). We conclude that mild replication stress sensitizes cells to the effect of taxol by favouring multipolar cell divisions, a process known to lead to cell death in cancer tissues.

## Discussion
Here we show that mild replication stress in non-cancerous cells provokes spindle architecture defects that are classically associated with chromosome gain/loss in cancer cells. Replication stress causes mis-segregation of entire or near-entire chromosomes via premature centriole disengagement and transient multipolar spindles. This disengagement depends on ATR, Cdk1

and Plk1 activity in G2, consistent with the idea that mild replication stress might deregulate the pathways controlling mitotic entry, resulting in an asynchrony between the DNA and the centrosome cycle. In addition, in a non-cancerous background mild replication stress may lead to higher microtubule stability, a condition that we find to favour premature centriole disengagement by itself. Replication-stress dependent premature centriole disengagement could be also observed in several CIN+ colorectal and breast cancer cell lines, implying that this mechanism may contribute to numerical aneuploidy in cancer tissues. Pre-cancerous lesions and cancer cells are frequently chromosomally instable, and chromosome gain or loss during mitosis is an important component of CIN, that results in numerical aneuploidy. Merotely due to spindle architecture defects or deregulated microtubule stability is the main mitotic dysfunction contributing to chromosome mis-segregation[5]. The different sources of these phenotypes, are however unclear: while multipolar spindles due to supernumerary centrosomes can be observed in some cancer tissues[53], their frequency is insufficient to explain the prevalence of numerical CIN observed in many different tumours and precancerous lesions[54]; mutations or deletions of mitotic genes that would explain the microtubule stability deregulation (e.g. microtubule depolymerases) are rare, most likely because these genes are essential at the cellular level[18]. Here, we show in non-cancer and cancer cells that mild replication stress may be one of the unknown causes underlying multipolar spindles (Fig. 6c). Transient multipolarity due to centrosome overduplication has long been shown to induce lagging chromosomes in anaphase[13,14]. Here we show that in RPE1 cells mild replication stress led to premature centriole disengagement in 20–25% of the cells (we suspect that the lower spatial resolution explains the lower rate seen in live cell imaging), and we found that transient multipolar spindles resulting from premature centriole disengagement lead to lagging DNA in 1/5 of the cases. This indicates that replication stress-induced premature centriole disengagement can lead to a chromosome mis-segregation rate that is comparable to the one seen in cancer cells (in the order of 5%), instead of the 0.5% found in RPE1 cells[40,55]. Moreover, it suggests that many more cancer cells might form transient multipolar spindles than just the subset of cells with extra centrosomes[53,56], and that drugs targeting spindle pole clustering might have a broader range of applications as originally anticipated. In addition our data indicate that mild replication stress also led to microtubule stabilization in a non-cancerous background. We find that this increased microtubule stability itself contributes to premature centriole disengagement, but it is likely to lead to segregation errors on its own by

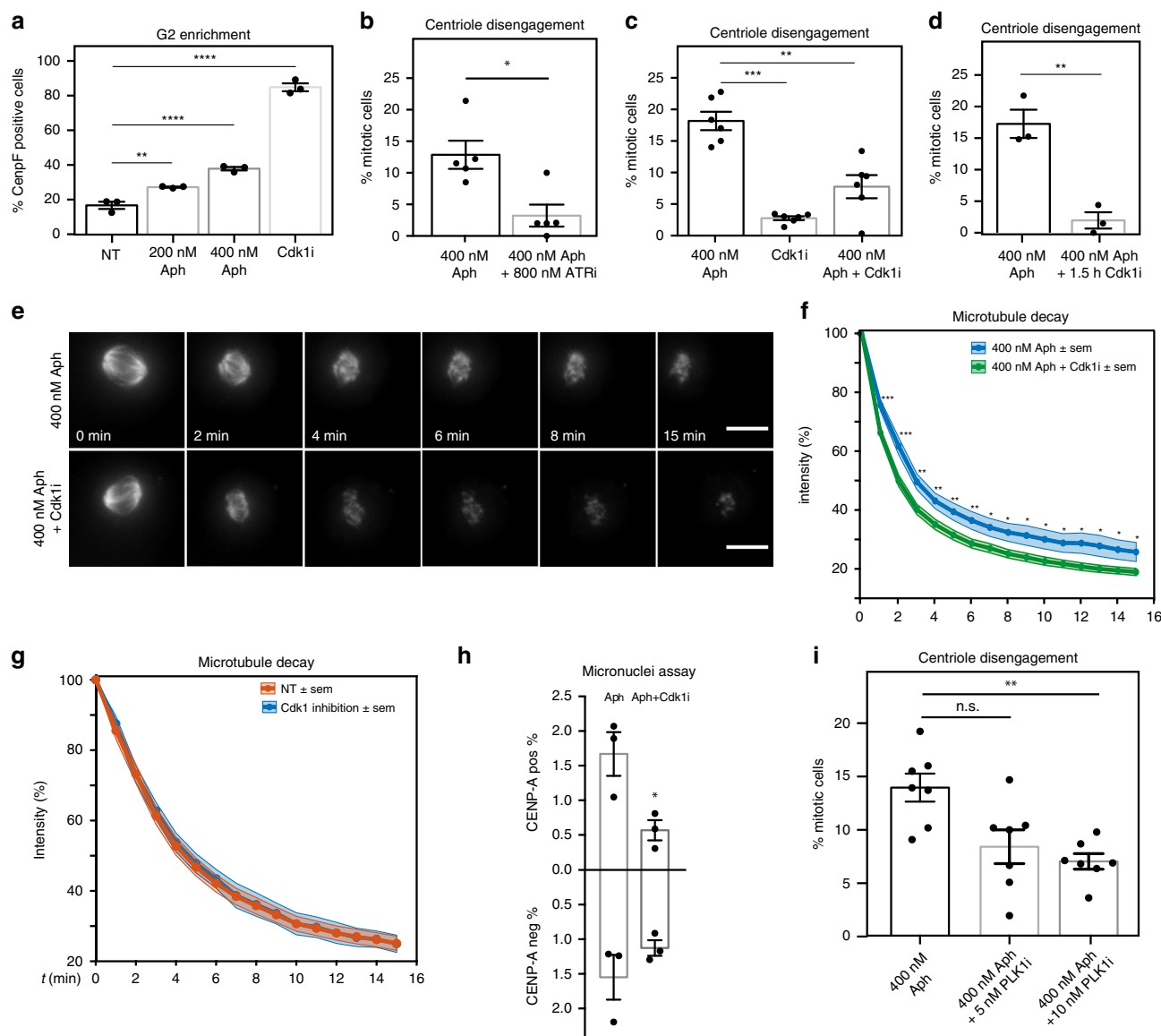

**Fig. 4** Replication stress phenotypes depend on Cdk1-, Plk1- and ATR. **a** Mean percentages of CENP-F positive RPE1 cells; $N = 3$ examining NT = 1398, 200 nM Aph = 1549, 400 nM Aph = 1632 and Cdk1i = 865 interphase cells; $p < 0.0001$ in a one-way Anova (NT vs. 200 nM Aph p = 0.005; NT vs. 400 nM Aph $p < 0.0001$; NT vs. Cdk1 inhibition $p < 0.0001$). **b** Mean percentages of disengaged RPE1 centrin1-GFP cells; $N = 5$ examining Aph = 210 and Aph/ATR = 221 mitoses; $p = 0.0275$ in paired $t$-test. **c** Mean percentages of disengaged RPE1 centrin1-GFP cells; $N = 6$ examining Aph = 369, Cdk1i = 440 and Aph/Cdk1i = 457 mitoses; $p < 0.0001$ in a one-way Anova (400 nM Aph vs. Cdk1 inhibition $p = 0.0005$ and 400 nM Aph vs. 400 nM Aph + Cdk1 inhibition $p = 0.003$). **d** Mean percentages of disengaged RPE1 centrin1-GFP cells; $N = 3$ examining Aph = 159 and Aph/Cdk1i = 208 mitoses; $p = 0.004$ in unpaired $t$-test. **e** Representative time-lapse images of metaphasic RPE centrin1-GFP cells stained with SiR-tubulin and treated with a 200 nM nocodazole pulse at $t = 0$ after indicated treatments; scale bars = 10 μm. **f** Quantification of the spindle signal as in Fig. 2d; $N = 20$ for Aph and 23 cells for Aph/Cdk1i; *$p < 0.05$; **$p < 0.01$; ***$p < 0.001$ in a repeated measurement Anova test. **g** Quantification of the spindle signal as in Fig. 2d; $N = 31$ for Cdk1i and 34 for NT. **h** Mean percentage of RPE cells containing CENP-A-negative and positive micronuclei. $N = 3$ examining Aph = 2421 and Aph/Cdk1i = 1793 interphase cells $p = 0.0339$ in unpaired $t$-test. **i** Mean percentages of disengaged RPE1 centrin1-GFP cells; N = 7 examining Aph = 321, Aph/5 nM Plk1i = 336 and Aph/10nMPlk1i = 357 mitoses; $p = 0.0377$ in a one-way Anova (400 nM Aph vs. 5 nM Plk1 inhibition $p = 0.1787$ and 400 nM Aph vs. 400 nM Aph + 10 nM PLK1 inhibition $p = 0.0091$). Error bars indicate sem. Source data are provided as a Source Data file

preventing correction of merotely, as seen for Kif2a/MCAK depletion[41]. Thus our results demonstrate how interphase stress can change mitotic parameters, illustrating how numerical CIN and structural CIN are in a complex relationship. Moreover, it provides a mechanism for the recent studies reporting that replication stress increases the incidence of whole lagging chromosomes[24,57,58].

Our findings are relevant in terms of cancer evolution since pre-cancerous lesions often show signs of DNA damage and replication stress[59]. Here we show that mild replication stress is sufficient to induce mis-segregation of entire or near-entire chromosomes in a non-cancerous background. We speculate that mild replication stress, which only leads to a transient G2 delay, might switch cell fate towards survival and genomic instability through such chromosome gain/loss. Since p53 dependent cell cycle checkpoints are not always activated by loss of single chromosomes[60], numerical aneuploidies could spread in a pre-cancerous population and/or release of DNA from micronuclei

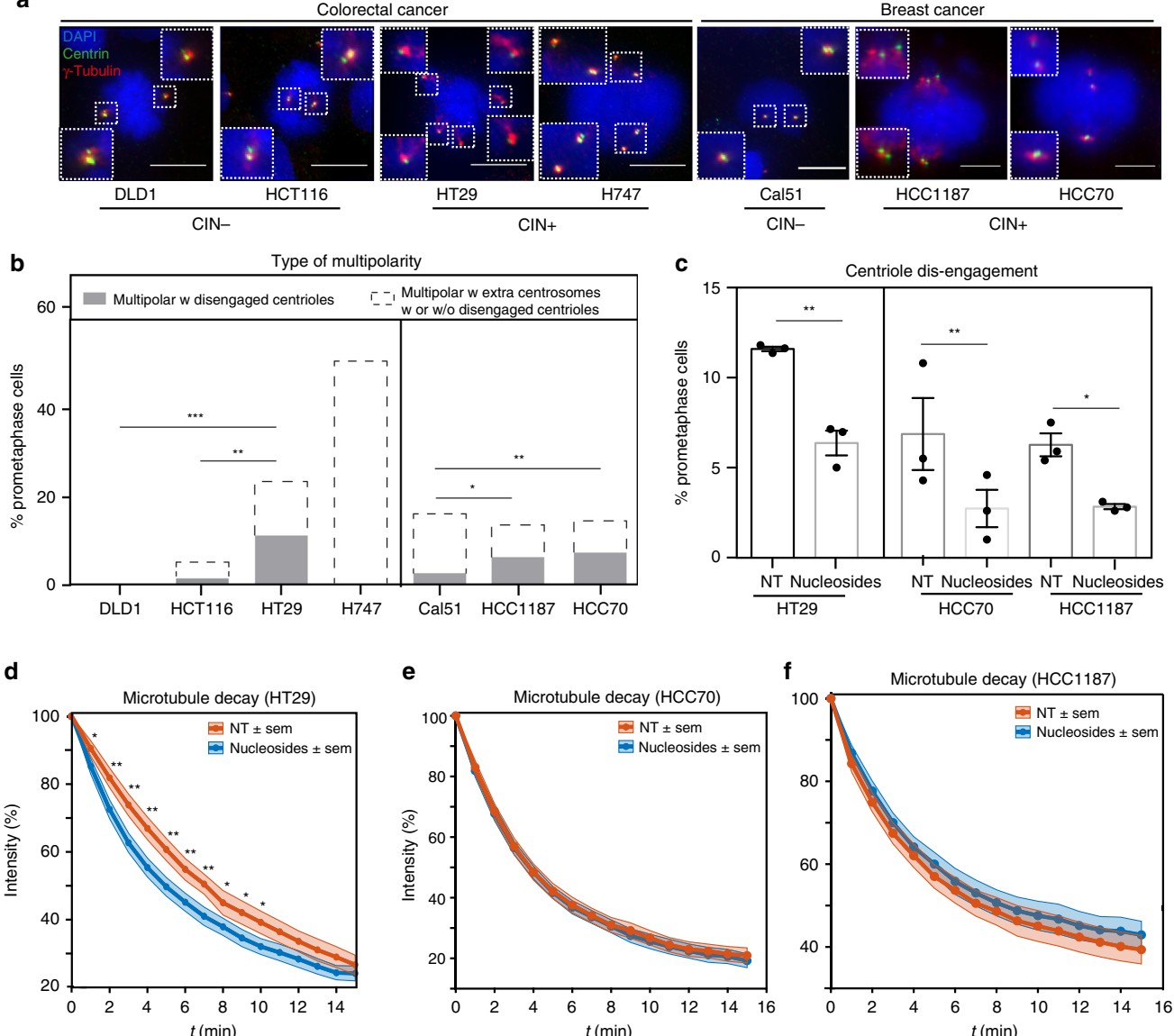

**Fig. 5** CIN cancer cells present centriole disengagement. **a** Representative images of prometaphase DLD1, HCT116, HT29, H747 Cal51, HCC70 and HCC1187 cells stained with antibodies against centrin (green), γ-tubulin (red) and DAPI (blue); scale bars = 10 μm. **b** Percentages of prometaphase cells with multipolar spindle with either premature centriole disengagement (grey) or extra centrosomes (white) in indicated cell lines; N = 93 prometaphase cells for DLD1, 92 for HCT116, 81 for HT29, 91 for H747, 203 for Cal51, 323 for HCC1187 and 359 for HCC70; p < 0.0001 in a chi-square test (DLD1 vs. HT29 p = 0.001, HCT116 vs. HT29 p = 0.0048, H7474 vs. HT29 p = 0.0011, Cal51 vs HCC1187 p = 0.0214 and Cal51 vs HCC70 p = 0.0066). **c** Mean per**c**entage of prometaphase HT29, HCC1187 and HCC70 cells with premature centriole disengagement with or without a 20 μM nucleosides supplement; N = 3 examining NT/HT29 = 138, nucleosides/HT29 = 125; NT/HCC1187 = 323, nucleosides/HCC1187 = 350, NT/HCC70 = 359 and nucleosides/HCC70 = 350 prometaphase cells; p < 0.0001 in two-way Anova test. (HT29 p = 0.002, HCC1187 p = 0.0158 and HCC70 p = 0.0065). **d–f** Quantification of the spindle intensity over time after a nocodazole pulse, as shown in Fig. 2d, in metaphase HT29 (**d**), HCC70 (**e**) or HCC1187 cells (**f**) supplemented with or without nucleosides; N = 33 for HT29/NT and 23 for HT29/nucleosides, 74 for HCC70/NT and 55 for HCC70/nucleosides, 74 for HCC1187/NT and 65 for HCC1187/nucleosides. *p < 0.05; **p < 0.01; ***p < 0.001 in a repeated measurement Anova test. Error bars indicate sem. Source data are provided as a Source Data file

could induce chronic inflammation[9]. Our data in CIN+ cancer cells suggests that replication stress can continue to drive chromosome mis-segregation in cancer cells via premature centriole disengagement. In contrast, microtubule stability did not appear to depend on replication stress in cancer cells, which could suggest that other mechanisms setting microtubule stability might become more prominent over time. Moreover, cancer cells tend to accumulate extra centrosomes over time[56], which may explain why pre-mature centriole disengagement has so far not been noticed in cancer cells.

Key future point of investigation will be to dissect the exact molecular mechanisms by which replication stress causes premature centriole disengagement. Our data indicate that microtubule stabilization can contribute to premature centriole disengagement, but that it is neither sufficient nor absolutely required. We conclude that replication stress must in addition affect the centrosome itself. This is consistent with previous studies, showing that inducible deletion of key DNA replication regulators can lead to centriole disengagement[61]. However, while previous studies speculated that centriole disengagement might

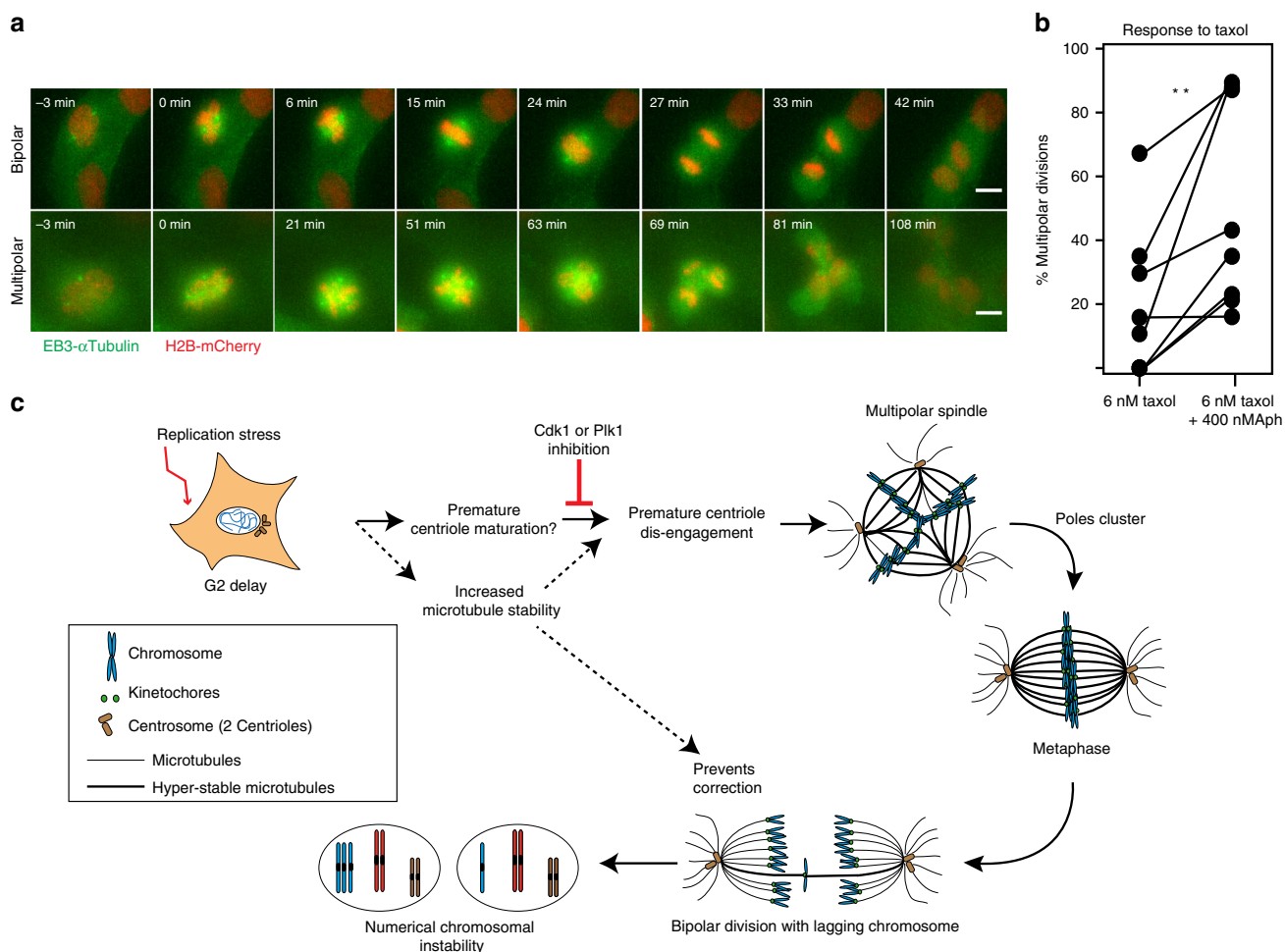

**Fig. 6** Cells with replication stress are more sensitive to taxol. **a** Representative time-lapse sequences of hTert-RPE1 H2B-mCherry/EB3-GFP cells dividing in a bipolar (top) or multipolar (bottom) fashion; scale bars = 10 μm. **b** Mean percentage of multipolar divisions in hTert-RPE1 cells treated with either 6 nM taxol alone or with 6 nM taxol + 400 nM Aphidicolin. Individual connected dots represent single paired experiments; $N = 8$ examining Taxol = 257 and Aph/Taxol = 214 anaphases; $p = 0.0097$ in paired $t$-test; Source data are provided as a Source Data file. **c** Speculative model on how mild replication stress impacts mitotic progression and leads to numerical aneuploidy; Mild replication stress imposed during S-phase may lead to premature centriole maturation. This induces a Cdk1 and Plk1-dependent premature centriole disengagement and causes transient multipolarity during the subsequent mitosis. These events elevate the incident of lagging chromosome in anaphase giving rise to daughter cells with unequal number of chromosomes. Note that in non-cancerous cells replication stress in addition increases microtubule stability, which may further favour the centriole dis-engagement and prevent correction of erroneous kinetochore-microtubule attachments in anaphase. Model created by T.W. and P.M.

result from long mitotic delays, our data suggest an earlier deregulation already in G2. Indeed, although, by immuno-fluorescence we do not see direct evidence for premature centriole disengagement in G2, the fact that centrioles split as soon as spindle microtubules pull on them in early prometaphase, and the fact that a pulse of Cdk1 inhibition in G2 can suppress the phenotype, indicates that mild replication stress must affect centrosomes in G2. Consistent, with this hypothesis, premature centriole disengagement depends on the ATR pathway, which can be activated by mild replication stress in G2. We envisage two non-exclusive possibilities. First, replication stress might partially disrupt the integrity of the peri-centriolar material (PCM), as reported for ionization radiation[62], facilitating premature disengagement as spindle microtubules pull on centrosomes after nuclear envelope breakdown. Consistent with this hypothesis, weakening PCM integrity can induce premature disengagement[63]. Alternatively, the prolonged G2 phase could lead to an excessive exposure to kinases that drive the centrosome cycle, priming mother centrioles for a new centriole duplication cycle by changing centrosome architecture[64], and thus disrupt the

synchronicity of the DNA and centrosome cycle. This might facilitate premature recruitment of PCM onto daughter centrioles, a step that normally takes place late in mitosis. It would allow all four centrioles to contact spindle microtubules too early, resulting in spindle pole fragmentation. This second hypothesis is supported by the fact that premature centriole disengagement depends on Cdk1 and its downstream target Plk1[49], which itself drives normal centriole disengagement during telophase[50]. It is also consistent with data in Drosophila spermatocytes, showing that Myt1 inhibition of Cdk1 in G2 prevents premature centriole disengagement[65]. Finally, it is consistent with our observation that disengaged centrioles always contain γ-tubulin from where microtubules emanate (Fig. 3).

Another point of future investigation will be to understand how replication stress might affect spindle microtubules in non-cancerous cells. Mitotic DNA damage can increase kinetochore microtubule stability via the mitotic kinases Plk1 and Aurora-A[55]. Since replication stress can lead to chromosome breakage in mitosis due to the endonucleolytic cleavage of under-replicated DNA[66], a similar signalling cascade could be involved.

Nevertheless, the fact that a transient inhibition of Cdk1 in G2 suppresses the higher microtubule stability rather indicates an origin in G2. Cdk1 has been previously shown to destabilize spindle microtubules in metaphase[48], whereas here we find that Cdk1 activity in G2 is required for increased microtubule stability, implying that Cdk1 activity might have differentiated effects on microtubule stability, depending on the cell cycle phase.

A final key aspect of our results is that replication stress sensitizes RPE1 cells to clinically relevant doses of taxol[35], by favouring multipolar cell divisions. This is remarkable, since taxol on its own leads to acentriolar extra spindle poles[35,42], indicating that different pathways leading to multipolar spindles can act in an additive manner. Whether these effects are due to the ability of mild replication stress to stabilize spindle microtubule or to induce premature centriole disengagement, remains however to be seen. Our data also provides a mechanistic explanation for the observation that mammary tumours treated with p38 inhibitors, which can lead to DNA stress, are more prone to respond to taxol[67]. This proof-of-principle experiment raise the possibility that replication stress might become a general biomarker for drugs targeting centrosome clustering.

## Methods

**Cell culture and, drug treatments**. HCT116, DLD1, HT29 and H747 cells (all kind gift of C. Swanton), as well as hTert-RPE1, hTert-RPE1 Centrin1-GFP (kind gift of A. Khodjakov) and hTert-RPE1 EB3-GFP/H2B-RFP (kind gift of W. Krek) cells were cultured at 37 °C in a 5% CO2 atmosphere in DMEM (Gibco) supplemented with 10% FCS (Labforce) and 1% penicillin/streptomycin mix (Gibco). Cal51 (kind gift of J. Curran), HCC70 (ATCC) and HCC1187 (ATCC) were cultured at 37 °C in a 5% $CO_2$ atmosphere in RPMI medium (Gibco). MCF10A (kind gift of J. Curran) were cultured at 37 °C in a 5% $CO_2$ atmosphere in RPMI-F12 medium (Sigma) supplemented with 10% FCS (Labforce), 1% penicillin/streptomycin mix (Gibco), 10 ng/ml EGF (sigma), 1 uM Dexamethasone (sigma) and 5ug/ml Insulin (sigma). All cell lines were genotyped using Short Tandem Repeats Profiling (Microsynth) and were tested for mycoplasma infection by PCR. Cells were treated with 200 nM, 400 nM (for RPE1) or 1 µM Aphidicolin (for MCF10A; for 16 h), 1 mM Hydroxyurea (for 3 h), 10 ng/ml Nocodazole (for 16 h), 9 µM RO3306, 5 nM or 10 nM Plk1 inhibitor (BI2536; Sigma), 800 nM ATR inhibitor (ETP-46464; Selleckchem) or 6 nM Taxol (added 3 h before the beginning of the movie, all Sigma Aldrich). For nucleoside rescue experiments in HT29, HCC70 and HCC1187 cells, a mix of deoxycytidine, deoxyadenosine, thymidine and deoxyguanosine (20 µM each, Sigma Aldrich) was applied for 48 h.

**RNA interference**. siRNA transfections were performed using lipofectamine RNAimax (Invitrogen) at a final concentration of 20 nM for each siRNA according to the manufacturer's instructions. The medium was replaced 24 h after the transfection and the microtubule decay assay was performed 48 h after the transfection. All applied siRNAs are based on sequences that have been previously validated: siControl (Qiagen, GGACCTGGAGGTCTGCTGT), siKif2a (Ambion, GTTGTTTACTTTCCACGAA), siMCAK (Dharmacon, GATCCAACGCAGTAATGGT)[68].

**Live cell imaging and analysis**. For live cell imaging experiments cells were imaged at 37 °C in Ibidi chambers (IBIDI) in L15 Leibovitz's medium supplemented with 10% FCS. To monitor cell division upon diverse treatments, cells were recorded every 3 min for 12 h using a 40x NA 1.3 oil objective on a Nikon Ti microscope equipped with a DAPI/FITC/Rhod/CY5 (Chroma, USA) filter set, Orca Flash 4.0 CMOS camera (Hamamatsu, Japan) and the NIS software. Z-stacks were imaged with z-slices separated by 2 µm, with 100 ms exposure per z-slice at wavelengths of 488 (525) and 561 nm (615 nm) excitation (emission). The time-lapse movies were analysed manually with NIS Elements software for mitotic timing, multipolarity, the presence of lagging chromosomes, or DNA bridges.

**Microtubule depolymerization assay**. To assess the stability of spindle microtubules in live conditions, cells were seeded in Ibidi chambers with L15 Leibovitz's + 10% FCS and incubated for 3 h with 25 nM SiR-Tubulin (Spirochrome) to label microtubules at steady-state concentrations. In case of cell lines overexpressing multidrug resistance pumps (HT29), cells were additionally treated with 10 µM Verapamil (Spirochrome). 30 min prior recording, cells were additionally treated with either 5 µM (for HT29, HCC70 and HCC1187) or 10 µM (for RPE1 and MCF10A cells) of MG132 (Sigma-Aldrich) to prevent mitotic exit. The Ibidi chambers were mounted on an Olympus DeltaVision microscope (GE Healthcare) equipped with a 60x NA 1.4 oil objective, a Cy5 filter set (Chroma Technology Corp) and a Coolsnap HQ2 CCD camera (Roper Scientific, USA) and maintained

at 37 °C. Mitotic cells were selected based on phase contrast microscopy, and spindle microtubule depolymerisation induced with a spike of 200 (RPE1 and MCF10A), 500 ng/ml (HT29), 400 ng/ml (HCC70) or 600 ng/ml (HCC1187) Nocodazole. Images were taken every minute for 15 min in the Cy5 channel to track microtubule depolymerization. The decay of fluorescence was measured using an automated home-made Matlab 2016a code (MathWorks Natick MA USA) that will be available under Github. Briefly, 4D images (xyzt) were first deconvolved using Softwork (GE Healthcare). For every single cell, segmentation relying on Otsu's method was done on a maximum intensity projection along the z-axis for all the time points using a custom-made framework developed with Matlab 2016a. The 4D images were summed along the z-axis generating a total value of the pixels assigned to the cell of interest. This allowed plotting the fluorescence intensity signal over time. Statistical analysis of obtained plots was done using repeated measures analysis of variance with Greenhouse-Geisser correction of the sphericity followed by a post-hoc multiple comparison by time.

**FACS analysis**. To assess the ability of Aphidicolin-treated cells to enter mitosis, cells were treated for 16 h with 0, 200 or 400 nM Aphidicolin and treated with 1000 nM nocodazole 3 h prior fixation (to trap cells in mitosis). Cells were pelleted, washed with PBS, fixed with 70% ice-cold ethanol, and stored at −20 ºC for at least 4 h. After a wash with PBT- buffer (0.1 g BSA and 5 µl Tween in 10 ml PBS), batches of $10^6$ cells were incubated for 20 min at RT with the mitotic marker Rat anti-Histone H3 tagged with AlexaFluor 647 (1/20, 558217, BD Bioscience). Cells were next washed with PBT buffer and stained with Propidium iodide /RNase staining buffer (550825, BD Bioscience). Labelled cells were detected with a Accuri C6 (BD) machine and analysed by FlowJo software (Tree Star).

**Metaphase spreads**. RPE cells were cultured for 16 h with 200 or 400 nM Aphidicolin and 1000 nM nocodazole 3 h prior collection. Cells were swollen in hypotonic solution (75 mM KCl and 15% FCS) for 10 min at 37 °C before fixation with an acetic acid solution (25% acetic acid, 75% ethanol). Fixed cells were spread onto glass cover slides and stained with Giemsa solution (32884, Sigma Aldrich).

**Immunofluorescence**. Cells were fixed 10 min in methanol at −20ºC, washed with PBS and blocked at RT in blocking solution (3% BSA in PBS), followed by staining with rabbit anti α-tubulin (1/500, ab1851, Abcam), rabbit anti γ-tubulin (1/2000; this study), mouse anti-CENP-F (1/1000, ab90, Abcam), mouse anti centrin (1/1000, 04–1624, Merck), γH2AX (1/2500, 05–636, Cell Signaling), 53BP1 (1/250, 4937, Cell Signaling) and CenpA antibodies (1/1000, ab13939, Abcam) in blocking solution. Cross-absorbed secondary anti-mouse and anti-rabbit antibodies (Invitrogen) were used. 3D image stacks of mitotic cells were acquired in 0.2 um steps using a 60x oil-immersion NA 1.4 objective on an Olympus DeltaVision microscope (GE Healthcare) equipped with a DAPI/FITC/Rhod/CY5 filter set (Chroma Technology Corp) and a CoolSNAP HQ camera (Roper-Scientific). The three-dimensional image stacks were deconvolved with SoftWorx (GE Healthcare). Image were cropped and processed with Fiji software. For the cold-stable assay cells were either treated with 400 nM Aphidicolin for 16 h or left untreated. Next, cells were treated with ice-cold medium for 12 min, washed once with Cytoskeleton buffer (10 mM MES, 150 mM NaCl, 5 mM EGTA, 5 mM Mgcl2, 5 mM Glucose, adjust pH at 6.1), fixed using a solution containing 0.1% glutaraldehyde, 8% formaldehyde and 0.1% Triton in Cytoskeleton buffer and stained for α-tubulin (1:1000, Sigma). To assess microtubule stability, the 3D images acquired were observed in 3D using Imaris software and cells were categorized into three different classes according to the abundance of kinetochore fibres. Single centrioles were considered disengaged when Centrin-GFP co-localized with γ-tubulin and the single centrioles were distant of more than 2 µm from each other.

**Antibody production**. Antibodies against γ-tubulin were raised by coupling the synthetic peptide CAATRPDYISWGTQEQ, corresponding to Cysteine plus the 15 C-terminal amino acids of human γ-tubulin, to keyhole limpet haemocyanin and injecting the coupling product in rabbits using a standard injection protocol (NeoMPS SA, France). Antibody sera were screened by immunofluorescence, and shown to recognize the typical γ-tubulin crescent at spindle poles, consistent with a previous study, which used exactly the same antigen.

**Statistical analysis**. Statistical analyses were performed using Prism 7.0 (GraphPad). All results were based on at least 3 independent experiments. Mean of averages are represented with s.e.m. errors bars and were evaluated with parametric tests (two-tailed t-test, One-way and Two-way Anova), categories based on classifications were evaluated with a Fischer's test. Differences were considered statistically significant when the P-value was < 0.05.

**Reporting summary**. Further information on research design is available in the Nature Research Reporting Summary linked to this article.

## Data availability

The primary and secondary data generated in the course of this project are available upon request. Due to their large size, they can only be sent on external hard disks. The source data underlying Figs. 1a, 1c, 1f, 1j, 2b, 2d, 2f, 2h, 2i 6d, 3b, 2, 3f, 3g, 3i, 4a-d, 4f-I, 5b-f and 6b and Supplementary Figs 1a and 2b are provided as a Source Data file.

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

## Acknowledgements

Authors are grateful to C. Swanton (Francis Crick Institute, London, UK), A. Khodjakov (Wadsworth Center Albany, New York, USA), W. Krek (ETH Zurich, Switzerland) and J. Curran (University of Geneva, Switzerland) for cell lines, D. Dudka and N. Liaudet (both University of Geneva, Switzerland) for advice on the microtubule depolymerisation assay, members of the Meraldi and Gotta laboratory for helpful discussions, and S. McClelland, I. Labidi-Galy, V. Naim, D. Dudka and L. Cirillo for the critical reading of the manuscript. Work in the Meraldi laboratory was supported by the Swiss National Science Foundation (SNF) project grants (No 31003A_160006 and 31003A_179413), the Swiss Cancer Research Foundation (grant KFS 3978–08–2016) and the University of Geneva. Therese Wilhelm was supported by a subsidy of the Ernest-Boninchi foundation and a Marie-Heim Vögtlein fellowship of the SNF.

## Author contributions

The project was initiated by T.W. and P.M. and directed by P.M. T.W. and A.M.O. carried out all the experiments. D.H. and F.S. contributed equally to this study: D.H. carried out immunofluorescence experiments for centriole disengagement and F.S. characterized the cancer cell lines; H.V. wrote the code for the microtubule depolymerisation assay; and A.E. produced the γ-tubulin antibody. T.W., A.M.O. and P.M. interpreted the data and wrote the manuscript with input of the other authors.

## Additional information

**Competing interests:** The authors declare no competing interests.

