## [Peer Review File · Nature Communications]

Reviewers' comments:

Reviewer #1 (Remarks to the Author):

NCOMMS-18-27306-T

Mild Replication Stress increases Microtubule Stability causing premature Centriole Disengagement and whole Chromosome Mis-segregation

Wilhelm et al demonstrate a novel role for replication stress in the stabilization of spindle microtubules and centriole disentanglement. While much is already known about the time played by cdk1 in both centriole biology and microtubule stability, the authors have demonstrated a mechanistic link between replication stress and numerical CIN.

However, these findings, while important, need to be more clearly controlled to clarify the novel role for replication stress in impacting mitotic progression and numerical aneuploidy. Particular concerns are listed below:

How general is this response to replication stress? The authors only demonstrate this with aphidicolin. Would other causes of replication stress induce the same phenomenon? The authors should address this with at least other drugs that induce replication stress or following oncogene exposure.

It is also unclear how general this phenomenon is - Most of the experiments were restricted to one non-cancerous cell model. The authors should expand the analysis to additional normal cells to observe how robust these responses are.

What exactly causes the microtubule stabilisation by replication stress? What other aspects of cdk1 signalling are implicated in their mechanism? Furthermore, have the authors used additional cdk1 inhibitors? To solidify the role of cdk1 in the replication stress induced microtubule stabilisation and premature centriole disengagement, the authors should consider the use of constitutively active cdk1 in their experiments.

Nucleoside supplementation only reduced microtubule stability in one of the CIN+ cells making it unlikely that this is a general mechanism of replication stress stabilising microtubules and would clearly need to be done in an extended panel of cell lines and using the live cell imaging assay.

The authors point out that the 'lagging DNA' observed after aphidicolin treatment could reflect whole or acentric chromosomes. They should look for lagging chromosomes in these anaphases using the centromeric marker to conclude whether it's a true mitotic defect and rule out the possibility that the observed segregation errors in reflect the presence of acentric chromosomes.

The degree of cold resistance phenotypes should be cell line dependent so the changes in microtubule stability should be compared to the untreated cell lines in each case and not relative to each other.

The increase in the extent of lagging chromosome formation amongst multipolar spindles vs bipolar spindles after aphidicolin treatment is potentially interesting but the error bars in fig 3C make it difficult to interpret. This experiment could benefit from being performed in additional cell lines with an increased number of anaphases.

The suppression of aphidicolin-induced chromosome mis-segregation by nocodazole treatment is also not very convincing. No quantification of untreated or nocodazole treated control cells are included making it difficult to ascertain the true extent of the rescue.

Reviewer #2 (Remarks to the Author):

In this manuscript, Wilhelm et al use the DNA polymerase inhibitor aphidicolin to induce replication stress in nontransformed, telomerase immortalized, human retinal pigment epithelial (RPE) cells. Aphidicolin increases the incidence of lagging chromosomes, most of which produce micronuclei. Lagging chromosomes are known to be caused by hyperstable kinetochore microtubules, and aphidicolin increases the cold stability of microtubules in the mitotic spindle. Lagging chromosomes are also caused by transient multipolar spindles, and aphidicolin-induced replication stress increases the incidence of transient multipolar spindles, which is predominantly due to premature centriole disengagement. CDK1 activity participates in aphidicolin-induced centriole disengagement and microtubule hyperstability, as both phenotypes were reduced by Cdk1 inhibition. In HT29 chromosomally unstable (CIN) colorectal cancer cells, which have hyperstable microtubules, microtubule stability was decreased by Cdk1 inhibition as well as nucleoside supplementation, which partially rescues replication stress. Aphidicolin also increased the incidence of multipolar spindles induced by low dose taxol treatment. The authors conclude that replication stress causes chromosome missegregation as a result of centriole disengagement and transient multipolar spindles, which result in lagging chromosomes and micronuclei.

CIN is common in human tumors, but mitotic genes are infrequently mutated in human cancer, so the causes of CIN in cancer remain unclear. Replication stress has been reported as one cause of CIN, although this is controversial (Bakhoum et al, Curr Biol 2014) and the mechanism remained unknown. Determining the causes of CIN in human cancer is of broad interest. The manuscript is well written and the data are well presented. The authors provide compelling evidence that aphidicolin can induce lagging chromosomes as a consequence of centriole disengagement and transient spindle multipolarity, which is an interesting phenotype. However, the emphasis on this phenotype seems to be misplaced since this appears to represent a relatively rare result of replication stress. This and a few other issues should be resolved prior to publication.

Major concerns

1. The manuscript emphasizes an interesting but apparently minor consequence of replication stress and represents it as the major consequence.
 - a. The incidence of lagging chromosomes is ~11% after treatment with 400 nM aphidicolin (figure 1f). ~7% of mitotic cells exhibit transient multipolarity after 400 nM aphidicolin (figure 3b), but only about 20% of the cells that display transient multipolarity have lagging chromosomes (figure 3c). This suggests that lagging chromosomes only occur in ~1.5% of cells due to transient spindle multipolarity and premature centriole disengagement, leaving the other ~9.5% to occur on bipolar spindles without centriole disengagement. This doesn't seem to support the conclusion that replication stress causes missegregation primarily via premature centriole disengagement and transient multipolar spindles (title, summary, page 9, twice on page 12).
 - b. The incidence of CENP-A (centromere) positive micronuclei is ~30% after treatment with 400nM aphidicolin, meaning ~70% of micronuclei contain chromosome fragments. It isn't clear the normalization to the monastrol treated cells is valid, but even if it is and 44% of micronuclei formed after 400 nM aphidicolin contain centromeres, more than half of micronuclei contain chromosome fragments. Since chromosome fragments are the most common type of CIN (structural CIN) caused by aphidicolin, they should at least be discussed. Also, since chromosome fragments represent the major type of chromosomal abnormality, they should be included in the quantitation of the metaphase spread data currently shown in figure 1c-d.
2. Most of the experiments are performed in a single cell type (RPE) using a single method to induce replication stress (aphidicolin). This raises questions about generalizability. Experiments in figure 5

test two CIN cancer cell lines, one of which shows hyperstable microtubules, replication stress, and some level of disengaged centrioles. Analysis of a wider panel of CIN cell lines would provide a better estimate of how commonly this occurs.

3. There seems to be quite a high rate of variability between experiments.

a. In the microtubule decay experiments, why is decay so much faster in the control in figure 2f than for the control in 2d? The change in the controls is larger than the effect caused by aphidicolin.

b. The percentage of cells with micronuclei after treatment with 400 nM aphidicolin ranged from 20-50% (figure 2g).

c. The percentage of cells with multipolar spindles after taxol treatment ranged from 0-70% (figure 6b).

4. The premature centriole disengagement finding is really interesting. The fact that it is dependent on Cdk1 activity is somewhat vague though. Is it dependent on PLK1 (which perhaps could be tested using PLK1 analog sensitive RPE cells or low doses of PLK1 inhibitor that permit spindle assembly as in Lera et al, JBC 2012? Or is it specific to cyclin A-CDK1 as opposed to cyclin B-CDK1 as in Dumitru et al, ELIFE 2017?

5. The experiments with taxol are interesting but somewhat underdeveloped. Does aphidicolin increase the number of spindle poles formed in response to taxol? To what extent does aphidicolin induce centriole disengagement in taxol? Does adding aphidicolin to taxol increase the level of cell death and reduce cellular viability?

Minor comments

6. The text (page 11) says H747 cells “showed multipolarity solely because of extra centrosomes” but figure 5g says it was solely because of disengaged centrioles. Which is it?

7. Why is the incidence of centriole disengagement (~26% in 3g, ~17% in 4b and 4c) so much higher than the incidence of transient multipolarity (~7% in 3b)? Shouldn't centriole disengagement cause transient spindle multipolarity?

8. The legend for figure S1 says that cells were treated with “16 hours 200 nM Aphidicolin + 3 hours nocodazole” and figure S1b-c indicate that cells are entering mitosis during this treatment. Was the aphidicolin washed out? Or are cells entering mitosis in the presence of 16 hours of 200 or 400 nM aphidicolin (which would mean that the movies of lagging chromosomes in figure 1 are likely of the 2nd mitosis after aphidicolin treatment rather than the first)?

9. The experimental design for the aphidicolin + Cdk1 inhibitor experiments in figure 4c is unclear. Was it 16 hour aphidicolin -> wash -> 1.5 hr RO-3306 -> wash? Or was the RO-3306 added during the last 1.5 hours of a 16 hour aphidicolin treatment?

10. In figure 1j, it would be helpful to know the extent of CENP-A positive micronuclei in untreated RPE1 cells. (The incidence of micronuclei in untreated RPE cells is low, but it should be relatively easy to scan thousands of interphase cells and only score those with micronuclei.)

11. Figure 3 shows that disengaged centrioles can organize microtubules but not necessarily that they can nucleate them, as stated on page 14.

12. Lagging chromosomes are certainly one cause of whole chromosome missegregation, but it is perhaps an overstatement to say that they are “the origin of whole chromosome gain/loss” (page 3).

13. It would be helpful when introducing SIR-tubulin to indicate that it is fluorescent docetaxel.

14. In figure 1h, what is the sample size?

15. In figure 2g, how long were the cells treated with aphidicolin or aphidicolin + nocodazole?

16. The "G2" peaks in figure S1c should be labeled "G2/M".

Reviewer #3 (Remarks to the Author):

Recent studies have shown that replication stress leads to chromosomal instability (CIN) through aneuploidy in addition to aberrations in chromosome structure, which is the more established mode for induction of CIN resulting from such stress. However, the mechanism for the former mode of CIN induction is not yet clear. In this manuscript, the authors show that induction of mild replication stress in normal diploid RPE cells by low dose Aphidicolin treatment promote microtubule stabilization causing multipolar spindles due to premature centriole disengagement, which in turn leads to lagging chromosomes during mitosis and micronuclei formation. Treatment with Nocodazole, which depolymerizes microtubules, reverses this effect. On the other hand, suppression of replication stress in cancer cells, which are inherently CIN+, was found to destabilize microtubules and prevent premature centriole disengagement. Finally, the study finds that mild replication stress enables the effect of Taxol in dividing cells by inducing multipolar mitoses. All in all, this is an elegant study which puts forward a mechanism by which replication stress can induce aneuploidy and is worthy of publication, pending that the authors address the following concerns:

(i) Fig. 1: It would be better to do these experiments in some sort of synchronized cells as that would give the authors better control over cell cycle progression. Is that not possible for RPE cells? Does Aphidicolin treated-cell show prolonged S phase or M phase? What is the phenotype of Hydroxyurea-treated cells as they go through S phase into M and through chromosome segregation? I think the authors should show that as a control. In Fig. 1g, for the sake of completeness it would be better to show control untreated cells.

For the purpose of the readers, it would be better to explain clearly what the DNA bridges and lagging chromosomes are in the context of producing CIN, i.e. what is the difference between them and the way they are produced?

(ii) Fig. 2: Class 3 defects observed could also be due to the fact that the spindle orientation is altered in response to the drug treatment. Is that the case here?

I really like the assay the authors used in Fig. 2c-f, but I also really feel that they should have done the 4 different conditions mentioned as one set of experiments (on the same day using the same imaging conditions) as this would give the reviewers/readers a better feel for comparing these results. Currently, it looks they have done these experiments separately. It would also make sense to merge the plots in Fig. 2d and f.

Personally, I am not very impressed with the difference observed with and w/o Nocodazole in Fig. 2g and was wondering why this is not more of a significant difference here based on the notion that this is a significant mechanism to produce chromosome mis-segregation as deemed by the authors. Please explain.

(iii) Fig. 3: I am really interested to know how the authors think microtubule stabilization (may it be replication stress of Taxol) produces multipolar spindles. Is the mechanism clearly known for Taxol? i.e., how does increased microtubule stability translate to acentriolar spindle poles?

Please show some sort of control in Fig. 3a at least to enable comparison of mitotic timing.

As the authors themselves conclude from their data, I think they need to tone down on the statement that increased MT-stability produced by Aphidicolin treatment is the main cause of lagging chromosomes as this distinction is observed even in untreated cells. I think the main conclusion should be that induction of multipolarity is important for mis-segregation, which is expected. Since there is increased multi-polarity after Aphidicolin treatment, it makes sense that there is more mis-segregation in this case.

Page 9 "but also by a mechanism independent of multipolar spindles." I would like the authors to substantiate on this statement to know their thoughts on this.

(iv) The results presented in Fig. 4 is only moderately interesting as the connection between G2/M transition-Cdk1 function and its role in centriole function/MT stability is reasonably well established. Based on the data presented in Fig. 3d-g and fig.4, I am interested to know if the centriole dis-engagement also happen in G2 well before NEBD unlike what the authors suggest in this study. Have the authors quantified as to how much the G2 timing is changed? (Hence the question earlier about use of synchronized cells). This also raises the concern that the replication stress induced by Aphidicolin might not be that "mild" after all.

v) Page 10, last sentence: please substantiate on "while other factors must be responsible for the elevated microtubule stability in H747 cells."

vi) In the experiments of Fig. 6a-b, it would have been good to have a sample size greater than 4. Please carry out more replicates to support the claim.

Minor points:

(i) Please check the grammar, in the last sentence of the 1st paragraph in page 11 beginning, "Importantly, both nucleoside.....". Specifically, what does "it" refer to in "strongly suggesting that it depends on"

(ii) Journal name not listed for reference # 18

Point-by-point response

Reviewer 1:

We appreciate that reviewer 1 thinks that our findings are novel and of general interest. We have addressed his/her concerns in the following manner

1. How general is this response to replication stress? The authors only demonstrate this with aphidicolin. Would other caused of replication stress induce the same phenomenon? The authors should address this with at least other drugs that induce replication stress or following oncogene exposure.

We address this concern in several ways:

- *First, we now demonstrate that centriole disengagement can be detected in 3 different colorectal cancer and breast cancer cell lines, and that this phenotype depends on endogenous replication stress as it can be suppressed by addition of nucleosides (new Fig. 5)*
- *Second, we demonstrate that Aphidicolin only induces centriole disengagement and microtubule stability when applied for a 16 hour period, but not when applied for a short period just before mitosis, arguing against an off-target effect (new Supplementary Fig. 2a&b)*
- *Third we show that centriole dis-engagement is rescued by partial inhibition of ATR, indicating that this phenotype depends on ATR signaling, arguing against an off-target effect of aphidicolin, that would be independent of replication stress (new Fig. 4b). These data provide also an additional mechanistic insight, demonstrating that centriole dis-engagement depend on the ATR signaling pathway.*
- *Fourth, as suggested by the reviewer we first attempted to transiently overexpress Cyclin E in RPE1 cells, an oncogene associated to replication stress; this led, however, to rapid cell death precluding any analysis of mitotic behavior.*
- *Finally, as an alternative drug we used Hydroxyurea, an inhibitor of nucleotide synthesis. Different doses of Hydroxyurea did not induce centriole disengagement, but rather led to a G1/S-arrest that depended on p53. Depletion of p53 in combination with a HU treatment nevertheless overcame this block and led to premature centriole disengagement after 48 hours. The HU treatment also led to very large cells and longer spindle length, implying different spindle mechanics that make any comparisons difficult. This suggests that HU and Aphidicolin induce different type of replication stress. Aphidicolin might impede complete replication at late replicating regions (either because of timing issues or R-loop formation at large genes that transcribe all over the cell cycle) whereas HU might rather provoke replication transcription conflicts earlier in S phase (early replicating fragile sites).*

Importantly, based on our analysis of the cancer cell lines, we believe that Aphidicolin recapitulates the centriole disengagement seen in cancer cells with replication stress. To not confuse the reader, we would prefer to not include the HU results in the manuscript, but make it available here to the reviewers, in case they believe that it is essential for the completeness of our story. Moreover, we note that our results are consistent with the known literature. As we now explicitly mention in our discussion, knock-down of several DNA replication factors by inducible CRISPR-Cas9 KO have been associated to centriole disengagement (McKinley and Cheeseman, Dev. Cell, 2017). While this study assumed that this phenotype might be the result of a long mitotic delay, we here directly show that

replication stress induces premature centriole disengagement without the need for a mitotic delay.

2. It is also unclear how general this phenomenon is - Most of the experiments were restricted to one non-cancerous cell model. The authors should expand the analysis to additional normal cells to observe how robust these responses are.

We thank reviewer 2 for this comment. To address this concern, we induced replication stress using Aphidicolin also in MCF10A cells, a non-tumorigenic breast epithelial cell line. In agreement with our results in RPE1 cells, Aphidicolin treatment increased both microtubule stability (Fig. 2) and the incidence of premature centriole disengagement (Fig. 3), suggesting that our findings are general for non-cancerous cell lines, and not only restricted to RPE1 cells.

3. What exactly causes the microtubule stabilisation by replication stress? What other aspects of cdk1 signalling are implicated in their mechanism? Furthermore, have the authors used additional cdk1 inhibitors? To solidify the role of cdk1 in the replication stress induced microtubule stabilisation and premature centriole disengagement, the authors should consider the use of constitutively active cdk1 in their experiments.

We agree with reviewer 1 that understanding how replication stress causes microtubule stabilization is a very interesting question, but feel that such experiments would go beyond the scope of revision experiments. In terms of Cdk1 inhibitor, to our knowledge RO-3306 is the most specific Cdk1 inhibitor available on the market, as all the other known inhibitors (e.g. roscovitin) also target additional Cdk kinases, which would prevent meaningful conclusions. Nevertheless, in our revised manuscript, we now also show that inhibiting Cdk1 in cells that did not experience replication stress has no effect on their microtubule stability, providing an additional specificity control. Testing a constitutively active Cdk1 mutant is not possible, as expression of such a mutant will immediately send cells directly from S-phase into cell division, resulting in a catastrophic mitosis with largely un-replicated DNA (Krek and Nigg, EMBO J, 1991).

Nevertheless, to expand our analysis we tested for the contribution of Plk1, a key downstream target of Cdk1 (Thomas et al., Cell reports, 2016). While it is not possible to fully inhibit Plk1 in Aphidicolin-treated cells, as Plk1 is essential for mitotic entry under those conditions (Macûrek L et al., Nature, 2008) and in general (Lilia Gheghiani et al., Cell reports, 2017), we now demonstrate that centriole disengagement is decreased by a partial chemical inhibition of Plk1. We thus speculate that Cdk1 might affect centrioles via Plk1, adding one additional mechanistic insight to our study.

4. Nucleoside supplementation only reduced microtubule stability in one of the CIN+ cells making it unlikely that this is a general mechanism of replication stress stabilising microtubules and would clearly need to be done in an extended panel of cell lines and using the live cell imaging assay.

As suggested by reviewer 1 we expanded our analysis of cancer cell lines by including also 2 CIN+ breast cancer cell lines that are prone to replication stress and one CIN- breast cancer cell line. First we show that these cell lines also display premature centriole dis-engagement in mitosis, confirming our results obtained with HT29. Second, we demonstrate that premature centriole dis-engagement can be decreased when endogenous replication stress is reduced by

supplementing growth media with nucleosides, This indicates that the endogenous replication stress seen in cancer cells induces premature centriole dis-engagement. However, we also find that nucleoside supplementation does not reduce microtubule stability in those additional 2 breast cancer cell lines, indicating that the result we obtained with HT29 is specific to this cell line.

Overall, this leads us to postulate that replication stress primarily leads to numerical CIN via premature centriole dis-engagement, a phenotype that we consistently found in all experiments and cell lines. Moreover, our results suggest that replication stress leads to microtubule stabilization in a non-cancerous background and possibly in few cancer cell lines. This is in line with our earlier postulate that microtubule stability in cancer cells might depend on different mechanisms that in some cases might include replication stress. Our results further suggest that microtubule stabilization enhances premature centriole dis-engagement in RPE1 cells, but that it is not essential (in contrast to Cdk1 activity), as a low nocodazole treatment only partially suppresses the phenotype. We have now also adjusted our discussion to better reflect these differentiated findings, and we thank reviewer 1 for suggesting these experiments.

5. The authors point out that the ‘lagging DNA’ observed after aphidicolin treatment could reflect whole or acentric chromosomes. They should look for lagging chromosomes in these anaphases using the centromeric marker to conclude whether it's a true mitotic defect and rule out the possibility that the observed segregation errors in reflect the presence of acentric chromosomes.

Given the scarcity of the lagging chromosomes and the fact that photo-toxicity often prevents to monitor the fate of single kinetochores in a large number of cells, it is very difficult to carry out such experiment and at the same time obtain sufficiently high number of events to be able to run a meaningful statistical analysis. For this reason we chose to monitor whole chromosome mis-segregation by counting the number of CENP-A positive micronuclei, which yields a much higher number of events.

6. The degree of cold resistance phenotypes should be cell line dependent so the changes in microtubule stability should be compared to the untreated cell lines in each case and not relative to each other.

We agree with the reviewer that our comparison was not useful. Since in our hands the microtubule depolymerization assay is more robust, we now only use this latter assay for the analysis of cancer cell lines in the revised Figure 5.

7. The increase in the extent of lagging chromosome formation amongst multipolar spindles vs bipolar spindles after aphidicolin treatment is potentially interesting but the error bars in fig 3C make it difficult to interpret. This experiment could benefit from being performed in additional cell lines with an increased number of anaphases.

First, we would like to point out that the obtained results are already statistically significant, despite the large variation seen from experiment to experiment. Second, these results are not particularly surprising, since there is ample published evidence indicating that the formation of a transient multipolar spindle will invariably result in a higher incidence of lagging chromosomes (e.g Ganem et al., Nature, 2009; Silkworth et al, Plos One, 2009). Given that these experiments are very time-consuming, that our results are significant and mostly confirm what has been previously observed, we feel that adding additional cell lines in this assay is beyond the scope of this revision.

8. The suppression of aphidicolin-induced chromosome mis-segregation by nocodazole treatment is also not very convincing. No quantification of untreated or nocodazole treated control cells are included making it difficult to ascertain the true extent of the rescue.

We agree with the reviewer and we now include in our revised manuscript the absolute percentages of CENP-A positive or negative micronuclei for all tested conditions, including in non-treated cells (Fig. 1, 2 and 5). This new representation gives a much better perspective on the effects of the different treatments, instead of the ratio's we used in the original manuscript.

Reviewer 2:

We are glad that reviewer 2 thought that our findings are compelling. We addressed his/her concerns in the following manner:

1. The manuscript emphasizes an interesting but apparently minor consequence of replication stress and represents it as the major consequence.
a. The incidence of lagging chromosomes is ~11% after treatment with 400 nM aphidicolin (figure 1f). ~7% of mitotic cells exhibit transient multipolarity after 400 nM aphidicolin (figure 3b), but only about 20% of the cells that display transient multipolarity have lagging chromosomes (figure 3c). This suggests that lagging chromosomes only occur in ~1.5% of cells due to transient spindle multipolarity and premature centriole disengagement, leaving the other ~9.5% to occur on bipolar spindles without centriole disengagement. This doesn't seem to support the conclusion that replication stress causes missegregation primarily via premature centriole disengagement and transient multipolar spindles (title, summary, page 9, twice on page 12).

We believe that this is a mis-understanding, as we should have explained those numbers in much more explicit manner in our discussion. Briefly based on live cell imaging we have 10.7% lagging DNA. Of those we estimate that more than half are real numerical CIN (Fig. 1j), which is the key question that we address in this manuscript (how can replication stress induce numerical CIN?). This gives us an incidence of 5% of lagging chromosomes, which is an order of magnitude more than the natural rate of chromosome mis-segregation observed in RPE1 cells (Kaseda et al, Biol Open, 2012). Based our experiments we have between 7% (live cell imaging, which may not detect dis-engaged centrioles that are still relatively close to each other) and 23% (fixed cell imaging) of cells with transient multipolar spindles. If the cells with such transient multipolar spindles have a lagging chromosome in 20% of the cases, we can estimate that the multipolar spindles will result in lagging chromosomes in up to 4.5% of all cell divisions. Thus we conclude that transient multipolar spindles are a major contributor to the numerical CIN seen after replication stress. Consistent with this hypothesis, we further show that suppression of centriole dis-engagement by Cdk1 inhibition also fully suppresses the appearance of CENP-A+ micronuclei in Aphidicolin-treated cells, while not affecting the rate of CENP-A- micronuclei. These points are now discussed explicitly in our revised discussion.

b. The incidence of CENP-A (centromere) positive micronuclei is ~30% after treatment with 400nM aphidicolin, meaning ~70% of micronuclei contain chromosome fragments. It isn't clear the normalization to the monastrol treated cells is valid, but even if it is and 44% of micronuclei formed after 400 nM aphidicolin contain centromeres, more than half of micronuclei contain chromosome fragments. Since chromosome fragments are the most common type of CIN (structural CIN) caused by aphidicolin, they should at least be

discussed. Also, since chromosome fragments represent the major type of chromosomal abnormality, they should be included in the quantification of the metaphase spread data currently shown in figure 1c-d.

We agree with reviewer that our representation of CENP-A positive and negative micronuclei based on ratios was confusing. As suggested by reviewer 1 we now use absolute values for all the micronuclei assays. This reveals that Aphidicolin increases the mis-segregation of both whole chromosome and fragments, that monastrol only increases the mis-segregation of whole chromosomes, and that nocodazole treatment or Cdk1 inhibition specifically suppresses the population of replication stressed induced CENP-A positive micronuclei.

2. Most of the experiments are performed in a single cell type (RPE) using a single method to induce replication stress (aphidicolin). This raises questions about generalizability. Experiments in figure 5 test two CIN cancer cell lines, one of which shows hyperstable microtubules, replication stress, and some level of disengaged centrioles. Analysis of a wider panel of CIN cell lines would provide a better estimate of how commonly this occurs.

We thank reviewer 2 for these suggestion. As detailed in our response to reviewer 1, we have now expanded our experiments to one additional non-cancerous cell line, and 3 additional cancer cell lines from breast tissues, and we find that in terms of premature centriole dis-engagement our results are general (see detailed response to reviewer 1)

3. There seems to be quite a high rate of variability between experiments.

a. In the microtubule decay experiments, why is decay so much faster in the control in figure 2f than for the control in 2d? The change in the controls is larger than the effect caused by aphidicolin.

b. The percentage of cells with micronuclei after treatment with 400 nM aphidicolin ranged from 20-50% (figure 2g).

c. The percentage of cells with multipolar spindles after taxol treatment ranged from 0-70% (figure 6b).

We agree with the reviewer that some of the experiments show a high degree of variability, which reflects the fact that we are working with living cells in which sometimes minor changes in the environmental conditions can lead to strong non-linear changes in the response. To deal with this day-to-day variability we have systematically carried out all our experiments as paired experiments on the same day with the exact same experimental conditions, and have focused our attention on the changes compared to control conditions.

a) this explains why the microtubule decay experiments are always shown as pairwise experiments

b) For the micronuclei experiments, we now plot the absolute values of the CENP-A- and CENP-A + micronuclei, which we find to give a more intuitive and more robust description of our findings (a ratio of two variable factors often shows high variability)

c) For the taxol experiments we only carried out paired experiments, while at the same time carrying out 8 independent experiments, which revealed that in almost every case Aphidicolin treatment resulted in a higher rate of multipolar cell divisions

4. The premature centriole disengagement finding is really interesting. The fact that it is dependent on Cdk1 activity is somewhat vague though. Is it dependent on Plk1 (which perhaps could be tested using PLK1 analog sensitive RPE cells or low doses of Plk1 inhibitor that permit spindle assembly as in Lera et al, JBC 2012? Or is it specific to cyclin A-CDK1 as opposed to cyclin B-CDK1 as in Dumitru et al, ELIFE 2017?

We thank reviewer 2 for this suggestion. We tested whether mild Plk1 inhibition would affect centriole dis-engagement in Aphidicolin-treated cells (strong Plk1 inhibition prevents mitotic entry in aphidicolin-treated cells) and found that the centriole disengagement phenotype was indeed partially rescued (new Fig. 4). We now also discuss our results in light of the results obtained by the Compton laboratory in Dumitru et al, eLife 2017. While the Dumitru study found that Cdk1/Cyclin-A destabilizes microtubules in metaphase, our experiments indicate that Cdk1-activity in G2 has a stabilizing influence on the future mitotic microtubules, indicating that we are not studying the same process

5. The experiments with taxol are interesting but somewhat underdeveloped. Does aphidicolin increase the number of spindle poles formed in response to taxol? To what extent does aphidicolin induce centriole disengagement in taxol? Does adding aphidicolin to taxol increase the level of cell death and reduce cellular viability?

We agree that the answers to these questions are potentially interesting, but we were unable to fully test them due to technical difficulties.

First, the presence of Aphidicolin did not increase the number of spindle poles in taxol-treated cells, as far as we can judge from our live-cell imaging resolution. Since taxol-treated cells often rotated in space, it is however, difficult to give precise numbers.

We also attempted to estimate whether taxol resulted in centriole dis-engagment, but found that taxol-treated cells contained few dis-engaged centrioles. However, one has to keep in mind that taxol also leads to a prolonged mitotic delay, and that dis-engaged centrioles might cluster during such a long time.

Finally, since Aphidicolin treatment substantially prolongs cell cycle duration, it is difficult to estimate whether cell viability is reduced, since all cell viability assays are influenced by cell proliferation speed. Aphidicolin will decrease the cell viability of taxol-treated cells, but based on the nature of the assay it is very difficult to judge whether this due to more mitotic errors or just a reduction in cell proliferation.

Nevertheless, to take in account these caveats we have now toned down our interpretation of the taxol results in the discussion.

Minor comments

6. The text (page 11) says H747 cells “showed multipolarity solely because of extra centrosomes” but figure 5g says it was solely because of disengaged centrioles. Which is it?

We thank reviewer 2 for this comment and we apologize for the mis-labeling of the figure 5g. Indeed, multipolarity in H747 cells depends only on the presence of extra centrosomes.

7. Why is the incidence of centriole disengagement (~26% in 3g, ~17% in 4b and 4c) so much higher than the incidence of transient multipolarity (~7% in 3b)? Shouldn't centriole disengagement cause transient spindle multipolarity?

As pointed out by the reviewer 2, premature centriole disengagement should induce transient multipolarity. However, this discrepancy between the two quantifications stems from the sensitivity of the two different methods used to quantify these values. To analyze the number of cells with disengaged centrioles, we used fixed cell imaging, where we stained for centrioles and centrosomes. Fixed cells are also imaged with higher z-stack resolution (one picture every 0.2um), allowing a very precise detection of number and position of centrioles. In live cell imaging; cells are imaged with a lower z resolution (one image every 2um) and with a time resolution of 3min to minimize

phototoxicity. Dis-engaged poles that cluster fast or are weak in intensity might therefore not be detected in live.

8. The legend for figure S1 says that cells were treated with “16 hours 200 nM Aphidicolin + 3 hours nocodazole” and figure S1b-c indicate that cells are entering mitosis during this treatment. Was the aphidicolin washed out? Or are cells entering mitosis in the presence of 16 hours of 200 or 400 nM aphidicolin (which would mean that the movies of lagging chromosomes in figure 1 are likely of the 2nd mitosis after aphidicolin treatment rather than the first)?

We apologize for the misunderstanding but cells treated with Aphidicolin are delayed in G2 compared to control cells but they eventually all proceed to mitosis (as seen in S1B and C with 400nM Aph). Aphidicolin is not washed out in any of the experiments as we work in unsynchronized conditions, where cells enter mitosis at very different time points during our 12 hour movies. RPE1 cells have a cell cycle timing of ~24h so during the 16 hours of treatment with Aphidicolin cells cannot perform two divisions. This point has now been clarified in the figure legend.

9. The experimental design for the aphidicolin + Cdk1 inhibitor experiments in figure 4c is unclear. Was it 16 hour aphidicolin -> wash -> 1.5 hr RO-3306 -> wash? Or was the RO-3306 added during the last 1.5 hours of a 16 hour aphidicolin treatment?

We thank reviewer 2 for this comment. In this experimental set up, we treated with Aphidicolin for 16h and we added Cdk1 inhibitor in the last 1.5h of the treatment. Before fixing the cells, we washed away the Cdk1 inhibitor to allow cells to enter mitosis. During all washes and the release we kept Aphidicolin in the media. We hope it is better explained in the revised version (corrected, page 11).

10. In figure 1j, it would be helpful to know the extent of CENP-A positive micronuclei in untreated RPE1 cells. (The incidence of micronuclei in untreated RPE cells is low, but it should be relatively easy to scan thousands of interphase cells and only score those with micronuclei.)

Thank you for this comment. We have now re-analyzed all the micronuclei quantifications and include now the number of CENP-A+ and CENP-A- micronuclei for non-treated cells in Fig. 1j.

11. Figure 3 shows that disengaged centrioles can organize microtubules but not necessarily that they can nucleate them, as stated on page 14.

We agree with the reviewer and have changed our formulation accordingly.

12. Lagging chromosomes are certainly one cause of whole chromosome mis-segregation, but it is perhaps an overstatement to say that they are “the origin of whole chromosome gain/loss” (page 3).

We now state that lagging chromosomes are the most frequent cause (but not the exclusive cause), consistent with the current literature in the field.

13. It would be helpful when introducing SIR-tubulin to indicate that it is fluorescent docetaxel.

Thank you for this comment. We added in the revised version that Sir-tubulin is a docetaxel-based fluorescent dye (corrected, page 8, 1st paragraph).

14. In figure 1h, what is the sample size?

The data presented in this graph refers to the live cell imaging analysis in Fig. 1f, so the sample size is $N = 3-6$; $n = 62-173$. This has been corrected in the figure legend of the revised manuscript.

15. In figure 2g, how long were the cells treated with aphidicolin or aphidicolin + nocodazole?

Cells were co-treated with Aphidicolin and nocodazole for 16 hours (corrected page 10, 2nd paragraph).

16. The “G2” peaks in figure S1c should be labeled “G2/M”.

Thank you for this comment. Indeed, this peak represents cells in G2 and M phase (corrected on the figure).

Reviewer 3

We appreciate that reviewer 3 thinks that our study is worth publishing if we can address the following concerns. Those points were addressed in the following manner.

1. Fig. 1: It would be better to do these experiments in some sort of synchronized cells as that would give the authors better control over cell cycle progression. Is that not possible for RPE cells? Does Aphidicolin treated-cell show prolonged S phase or M phase? What is the phenotype of Hydroxyurea-treated cells as they go through S phase into M and through chromosome segregation? I think the authors should show that as a control. In Fig. 1g, for the sake of completeness it would be better to show control untreated cells.

With regard to cell synchronization, as we explained in the manuscript we avoided any synchronization procedure, as such methods invariably lead to additional stress (aphidicolin, thymidine or Hydroxyurea block) or even chromosomal loss (nocodazole release) on its own, which would prevent meaningful conclusions. We show in our manuscript (Fig. 4A and 3F) that low doses of aphidicolin lead to a prolonged G2 phase, but do not prolong M-Phase. As stated in point 1 of reviewer 1, we also tested the effects of Hydroxyurea, which we found to be complex: first in normal RPE1 cells it leads to a p53-dependent arrest at the G1/S transition; in cells lacking p53 it does lead to centriole disengagement, but only after 48 hours. At this stage, these cells are larger and have also larger spindles, which in our hands show differences in terms of spindle mechanics, which is why we prefer not to include these data in this current study.

For the purpose of the readers, it would be better to explain clearly what the DNA bridges and lagging chromosomes are in the context of producing CIN, i.e. what is the difference between them and the way they are produced?

We now introduce the concept of DNA bridges and lagging chromosome more explicitly in the first paragraph of the introduction. We also explain how we differentiate between the two phenotypes when analyzing live cell imaging movies in the results section (page 6).

(ii) Fig. 2: Class 3 defects observed could also be due to the fact that the spindle orientation is altered in response to the drug treatment. Is that the case here?

For the analysis of the cold-treated cells, all images are recorded as 3D stacks and evaluated in 3D using Imaris, which allows us to analyze Class 3 types of cells irrespective their orientation. This information is now explicitly stated in the Material and Methods section.

I really like the assay the authors used in Fig. 2c-f, but I also really feel that they should have done the 4 different conditions mentioned as one set of experiments (on the same day using the same imaging conditions) as this would give the reviewers/readers a better feel for comparing these results. Currently, it looks they have done these experiments separately. It would also make sense to merge the plots in Fig. 2d and f.

As concluded by the reviewer, the experiments shown in Figure 2c/d and 2e/f were indeed done on different days, as the recording of a whole experimental series of single cells takes 4 hours. This means that we can in one day reasonably compare two different conditions (e.g. +/- replication stress) making sure that the cells are at the exactly same temperature, on the same dish with parallel chambers and incubated in the same nocodazole-containing medium. However, expanding this to more conditions is not feasible in one working day, which is why we have to treat them as separate experiments.

Personally, I am not very impressed with the difference observed with and w/o Nocodazole in Fig. 2g and was wondering why this is not more of a significant difference here based on the notion that this is a significant mechanism to produce chromosome mis-segregation as deemed by the authors. Please explain.

We thank reviewer 3 for his/her comment. We now have re-analyzed all the micronuclei quantification experiments, plotting the absolute numbers of CENP-A+ and CENP-A- micronuclei, which reveals a much more robust picture. As shown in the novel Fig. 2i, low doses of nocodazole efficiently rescue the incidence of CENP-A+ micronuclei, while leaving the CENP-A- micronuclei unaffected.

(iii) Fig. 3: I am really interested to know how the authors think microtubule stabilization (may it be replication stress of Taxol) produces multipolar spindles. Is the mechanism clearly known for Taxol? i.e., how does increased microtubule stability translate to acentriolar spindle poles?

Taxol has been shown to cause the formation of extra acentriolar poles and multipolar divisions in clinically relevant doses (within the range we use for our analysis), leading to highly aneuploid daughter cells that will die in the subsequent interphase (Zasadil et al., science translational medicine, 2014). Jessica E Hornick and colleagues have partially explained how these extra acentriolar poles are formed; when taxol-treated cells enter mitosis, microtubules are re-distributed from the centrosomes to the cortex where they recruit HSET and Numa and finally form asters and multipolar spindles. Overall, authors claim that these cortical poles form due to the inability of taxol-treated cells to rapidly transport different factors, such as Numa, from the cortex to the centrosomes (Hornick et al., Cell motil. Cytoskeleton, 2008).

Please show some sort of control in Fig. 3a at least to enable comparison of mitotic timing.

We apologize for the mis-understanding, but mitotic timing in cells with and without replication stress is already shown in the current Figure 3f.

As the authors themselves conclude from their data, I think they need to tone down on the statement that increased MT-stability produced by Aphidicolin treatment is the main cause

of lagging chromosomes as this distinction is observed even in untreated cells. I think the main conclusion should be that induction of multipolarity is important for mis-segregation, which is expected. Since there is increased multi-polarity after Aphidicolin treatment, it makes sense that there is more mis-segregation in this case.

We agree with reviewer 3 that the multipolar spindles are likely to be the major source of lagging chromosomes as stated in point 1 of reviewer 2. In light of our new results obtained with a panel of cancer cell lines we now put a much stronger emphasis on the role of pre-mature centriole dis-engagement in generating numerical CIN. Nevertheless, based on our old and new experiments in MCF10A we still believe that replication stress also induces microtubule stabilization in non-cancerous cells, and that this phenotype contributes to centriole dis-engagement. We hope that our revised discussion better reflects this new emphasis.

Page 9 “but also by a mechanism independent of multipolar spindles.” I would like the authors to substantiate on this statement to know their thoughts on this.

Please, see point above.

(iv) The results presented in Fig. 4 is only moderately interesting as the connection between G2/M transition-Cdk1 function and its role in centriole function/MT stability is reasonably well established. Based on the data presented in Fig. 3d-g and fig.4, I am interested to know if the centriole dis-engagement also happen in G2 well before NEBD unlike what the authors suggest in this study. Have the authors quantified as to how much the G2 timing is changed? (Hence the question earlier about use of synchronized cells). This also raises the concern that the replication stress induced by Aphidicolin might not be that “mild” after all.

First, we kindly disagree that the role of Cdk1 in linking centriole function and MT stability is reasonably well established; as to our knowledge our results are novel and have not been described before. Indeed, the main link between Cdk1 and MT stability stems from the Compton laboratory (Dimitru et al., 2017) which showed that Cdk1 has a microtubule destabilizing activity in metaphase, whereas our results point to a microtubule stabilizing activity that derives from the Cdk1 activity in G2.

Second, we have now extended our analysis of the signaling pathways controlling centriole disengagement, and shown that it depends on ATR and Plk1.

Third, we now state explicitly that centriole dis-engagement was not yet visible in G2. Indeed as state in our text, we only saw dis-engagement once cells entered mitosis, presumably due to pulling forces exerted by the spindle microtubules

Fourth, as stated in our results section, Aphidicolin treatment led to a G2 delay. We further show that this delay is not sufficient for a pre-mature centriole dis-engagement, as an equivalent Cdk1 inhibition and release does not lead to premature centriole dis-engagement.

v) Page 10, last sentence: please substantiate on “while other factors must be responsible for the elevated microtubule stability in H747 cells.”

This sentence is now not part any more of the manuscript, since we do not discuss the microtubule stability of H747 cells. Nevertheless, we now emphasize that premature centriole dis-engagement is the phenotype that is common to all our experiments. Microtubule stabilization might play a role in a non-cancerous setting, but is not general to the replication stress seen in cancer cell lines.

vi) In the experiments of Fig. 6a-b, it would have been good to have a sample size greater than 4. Please carry out more replicates to support the claim.

First, we would like to point out that we have already carried out 5 paired experiments as explained under point 3 of reviewer 2, which all showed the same qualitative response: i.e. an increase in multipolar divisions after aphidicolin co-treatment and the difference was already significant. Nevertheless, to address the concern of the reviewer, we include now 3 more paired experiments, which make the significance of this result even greater.

Minor points:

(i) Please check the grammar, in the last sentence of the 1st paragraph in page 11 beginning, “Importantly, both nucleoside.....”. Specifically, what does “it” refer to in “strongly suggesting that it depends on”

We thank the reviewer for pointing out this issue. This sentence has been removed from the manuscript during the revision of the text.

(ii) Journal name not listed for reference # 18

We thank the reviewer. We corrected this in the revised version.

Figure for reviewers; Wilhelm et al.

Supplementary Figure for reviewer. Mild doses of Hydroxyuread also lead to premature centriole disengagement in p53-depleted cells (a) FACS cell cycle profile of RPE1 cells treated for either for 3 hours with 1 μ M nocodazole only to trap mitotic cells or for 16 hours 300 μ M Hydroxyurea (HU) + 3 hours 1 μ M nocodazole. Cells were sorted according to their DNA content (propidium iodide staining) and the mitotic marker phospho-H3. Unlike Aphidicolin, mild HU treatment led to a strong G1/S arrest and barely any mitotic cells. This indicated that replication stress induced by HU affects the cell cycle differently. (b and c) To avoid the G1/S arrest induced by HU RPE+ cells were depleted of p53, which allowed them to progress through the cell cycle. Under such conditions cells were delayed in the cell cycle but nevertheless entered mitosis. We found that a 48 hours HU treatment resulted in a strong centriole dis-engagement phenotype. To quantify this phenotype cells were stained with Centrin-1 (green; centrioles) and γ -tubulin antibodies (red; PCM), as well as DAPI (b). Quantification of 3 independent experiments indicated higher rate of centriole dis-engagement in HU-treated cells (c). Nevertheless, since the nature of the cycle arrest, the status of p53, the duration of the treatment, and the size of the spindle after the treatment strongly differ from our Aphidicolin treatment, we would prefer not including those results, to avoid an unnecessary confusion.

REVIEWERS' COMMENTS:

Reviewer #1 (Remarks to the Author):

I am satisfied with the revisions made to the manuscript

Reviewer #2 (Remarks to the Author):

The revised manuscript from Wilhelm et al is substantially improved. It convincingly demonstrates the claims made in the title and summary, specifically that mild replication stress causes centriole disengagement and chromosome missegregation. Several minor points remain to be addressed.

1. The text on page 12 says, "an equal number of multipolar spindles displayed a combination of disengaged centrioles and overduplicated centrosomes (Fig. 5b)", but figure 5b doesn't appear to show this. Are the dashed boxes meant to indicate spindles that have both disengaged and overduplicated centrosomes (rather than just overduplicated centrosomes as currently indicated in the legend)?
2. In figure S2b, the bar for 400nM Aph (1.5 hrs) is missing. If the multipolar spindles don't show centriole disengagement, the bar should be gray. Also, why does the bar for 400 nM Aph (16 hrs) only quantify ~25% of the multipolar spindles? Is the y axis meant to be labeled "% mitosis" rather than "% multipolar mitosis"?
3. There is a disconnect between the text on page 9, which says "In >90% of the Aphidicolin-treated RPE1-cells with multipolar spindles we found single centrioles surrounded by γ -tubulin in the extra spindle poles" and figure 3d, which shows ~8-22%. Is the y axis meant to be labeled "% mitosis" rather than "% multipolar mitosis"?
4. The inset is missing in the "Overduplication" panel in figure 3h.
5. Insets are necessary in the images in figure 5a to show whether centrosomes contain 1 or 2 centrioles.
6. In the abstract, the word "stress" is missing at the end of the 11th line (...cancer cell lines with endogenous replication [stress] and that...)

Reviewer #3 (Remarks to the Author):

In the resubmitted manuscript, the authors are more reasonable in some of their assertions, which in turn are better backed by supporting data. Also, substantial amount of additional data using multiple cell lines has been added to the manuscript to address the reviewer's concerns. However few minor concerns remain which are listed below:

- (i) I had asked to show control untreated cells obtained under similar conditions in Fig. 1g, which the authors have ignored.
- (ii) I do not understand why the authors cannot provide stills from control movies without Aph treatment in Fig. 3a. From the plot in Fig. 3f, is it possible there is a slight delay in anaphase onset at 400nM Aph?
- (iii) I agree with the concern in point # 7 raised by reviewer 1 with regard to the lagging

chromosomes. If it is that the experiments are time consuming, maybe the assay can be performed in at least one if not two additional cell lines.

Point-by-point response:

Reviewer #1 (Remarks to the Author):

I am satisfied with the revisions made to the manuscript

We thank the reviewer for the positive assessment.

Reviewer #2 (Remarks to the Author):

The revised manuscript from Wilhelm et al is substantially improved. It convincingly demonstrates the claims made in the title and summary, specifically that mild replication stress causes centriole disengagement and chromosome missegregation. Several minor points remain to be addressed.

We thank the reviewer for the positive assessment and have addressed his/her remaining comments in the following manner:

1. The text on page 12 says, "an equal number of multipolar spindles displayed a combination of disengaged centrioles and overduplicated centrosomes (Fig. 5b)", but figure 5b doesn't appear to show this. Are the dashed boxes meant to indicate spindles that have both disengaged and overduplicated centrosomes (rather than just overduplicated centrosomes as currently indicated in the legend)?

We thank reviewer 2 for his comment. The dashed boxes indeed contained cells with overduplicated centrosomes concomitant with or without disengaged centrioles. Since these categories were too complex for a clean analysis, they were excluded from further analysis. We have, nevertheless adapted the Figure legend in Figure 5b to avoid this confusion.

2. In figure S2b, the bar for 400nM Aph (1.5 hrs) is missing. If the multipolar spindles don't show centriole disengagement, the bar should be gray. Also, why does the bar for 400 nM Aph (16 hrs) only quantify ~25% of the multipolar spindles? Is the y axis meant to be labeled "% mitosis" rather than "% multipolar mitosis"?

First, we indeed mis-labelled the y-axis and corrected this error for Fig S2B, Fig3B and Fig3D. Second, the bar for 400nM Aph (1.5 hrs) was not missing, there were just 0% of the cells with dis-engaged centrioles. Since we have now included dots for the averages of each experiment, this point should now be clear.

3. There is a disconnect between the text on page 9, which says "In >90% of the Aphidicolin-treated RPE1-cells with multipolar spindles we found single centrioles surrounded by γ -tubulin in the extra spindle poles" and figure 3d, which shows ~8-22%. Is the y axis meant to be labeled "% mitosis" rather than "% multipolar mitosis"?

We thank the reviewer for pointing this out. As mentioned in the previous paragraph this has been now corrected.

4. The inset is missing in the "Overduplication" panel in figure 3h.

These insets are now added in the revised version of the manuscript.

5. Insets are necessary in the images in figure 5a to show whether centrosomes contain 1 or 2 centrioles.

Insets are now added for Fig5a.

6. In the abstract, the word “stress” is missing at the end of the 11th line (..cancer cell lines with endogenous replication [stress] and that...)

We thank the reviewer for pointing this out. The missing word was added.

Reviewer #3 (Remarks to the Author):

In the resubmitted manuscript, the authors are more reasonable in some of their assertions, which in turn are better backed by supporting data. Also, substantial amount of additional data using multiple cell lines has been added to the manuscript to address the reviewer’s concerns. However few minor concerns remain which are listed below:

(i) I had asked to show control untreated cells obtained under similar conditions in Fig. 1g, which the authors have ignored.

We have now, as requested, added a sequence of stills from live cell imaging of control untreated cells in Figure 1g in the revised version of the manuscript.

(ii) I do not understand why the authors cannot provide stills from control movies without Aph treatment in Fig. 3a. From the plot in Fig. 3f, is it possible there is a slight delay in anaphase onset at 400nM Aph?

First, we have now added stills from movies of untreated control cells showing a normal bipolar spindle in Figure 3a.

Second, yes there is a small delay in the median mitotic timing of 3mins that becomes statistically significant, due to the very high number of measured cells. However, we do not believe that this small delay is biologically significant. Indeed, Karki and colleagues (Nat. Comm, 2017) had to retain cells in mitosis for at least 2 hours to observe a similar incidence of premature centriole dis-engagement. Nevertheless, to better reflect our findings we now state in the text

“Live cell imaging showed that mild replication stress did not change median mitotic timing by more than 3 minutes (Fig. 3f), which indicated that centriole disengagement was not caused by a prolonged mitotic duration, unlike what has been seen in other conditions⁴⁴.”

(iii) I agree with the concern in point # 7 raised by reviewer 1 with regard to the lagging chromosomes. If it is that the experiments are time consuming, maybe the assay can be performed in at least one if not two additional cell lines.

As stated in our previous point-by-point reponse, the fact that multipolar spindles tend to lead to lagging chromosomes has been well documented by the initial studies in the Pellman and Cimini laboratories (e.g Ganem et al., Nature, 2009; Silkworth et al, Plos One, 2009), and confirmed in numerous follow-up studies. We believe that this specific aspect of our study (multipolar spindles leading to lagging DNA), which is in full agreement with the consensus in the field, is our least surprising result. Moreover, repeating such experiments in other cell lines would require the generation of new stable cell line, the validation of those cell lines in terms of chromosome

segregation accuracy, and very time-consuming live-cell experiments that would take months. We therefore reasonably feel that such experiments are beyond a normal revision cycle, and not required since it would only confirm something that is already known.